# TBXA2R activates ERMs to drive motility, invasion, and metastatic colonization of TNBC cells

Kévin Leguay[1,2,*] , Omaima Naffati[1,2,*], Camila Lie Kiyan[1,2,3], Yu Yan He[1,2,3], Mireille Hogue[1,3], Chloé Tesnière[1,4], Melania Gombos[1,4], Hellen Kuasne[10], Louis Gaboury[1,5,6,7], Christian Le Gouill[1,3], Sylvain Meloche[1,4,6,8] , Michel Bouvier[1,3,6,9] , Sébastien Carréno[1,2,6,7]

**Cell migration and invasion are critical for cancer cell metastasis, relying on the ability of cells to adapt their morphology. Proteins of the ezrin, radixin, and moesin (ERM) family are key regulators of cell morphogenesis and essential determinants of cancer cell metastasis. However, the mechanisms by which ERMs are activated in metastatic cells remain poorly understood. We identified the thromboxane A2 receptor (TBXA2R), a G protein–coupled receptor overexpressed in multiple cancers, as a critical activator of ERMs, enhancing the motility and invasion of triple-negative breast cancer cells. We found that TBXA2R activates ERMs by engaging the $G\alpha_{q/11}$ and $G\alpha_{12/13}$ subfamilies, the Rho subfamily of Rho GTPases, and their Ser/Thr kinase effectors SLK and LOK. Furthermore, we demonstrate that TBXA2R promotes triple-negative breast cancer cell motility and invasion in vitro and metastatic colonization in vivo, dependent on ERM function. This reveals a novel signaling axis by which a member of the largest class of receptors activates key metastatic determinants, thereby controlling various aspects of metastasis. This discovery opens new avenues for developing targeted therapies against cancer metastasis.**

## Introduction

Ezrin, radixin, and moesin (ERMs) are a family of membrane–cytoskeleton linker proteins that regulate cell morphogenesis, migration, and invasion (Fehon et al, 2010). They integrate actomyosin forces and microtubule signaling at the cell cortex by cross-linking actin filaments (Algrain et al, 1993) and microtubules (Solinet et al, 2013) with the plasma membrane. Inactive ERMs cycle between the cytoplasm and the plasma membrane in a closed conformation, where their N-terminal FERM domain binds

their C-terminal C-ERMAD domain (Leguay et al, 2021). Upon stimulation, ERMs open up and expose their FERM microtubule-binding domain (Solinet et al, 2013) and their C-ERMAD actin-binding site (Gary & Bretscher, 1995). The C-ERMAD also harbors a conserved regulatory threonine residue (T567, T564, and T558 in ezrin, radixin, and moesin, respectively) that is phosphorylated upon stimulation to maintain the active open conformation (Simons et al, 1998). ERMs were shown to control cell morphogenesis during mitosis (Carreno et al, 2008; Kunda et al, 2008; Roubinet et al, 2011; De Jamblinne et al, 2020; Leguay et al, 2022), cell migration (Arpin et al, 2011; Hoskin et al, 2015; Barik et al, 2022), and cell invasion (Estecha et al, 2009; Song et al, 2020). In a pathological context, high levels of ezrin, radixin, or moesin expression correlate with a high rate of metastasis and poor prognosis in several cancers (Clucas & Valderrama, 2014). For instance, increased expression of moesin correlates with metastatic progression of oral squamous cell carcinoma (Kobayashi et al, 2004), melanoma (Estecha et al, 2009), and breast carcinoma (Bartova et al, 2017). Radixin is overexpressed in prostate cancer (Bartholow et al, 2011). Ezrin is overexpressed and hyperactivated in several cancer cells, including metastatic rhabdomyosarcoma (Yu et al, 2004), osteosarcoma (Khanna et al, 2004), prostate cancer (Pang et al, 2004), and mammary carcinoma (Bruce et al, 2007). In mice, ERM inhibition prevents breast cancer or osteosarcoma metastasis (Khanna et al, 2004; Elliott et al, 2005). These examples and other observations led to the notion that ERMs play central roles in metastasis (Curto & McClatchey, 2004; Barik et al, 2022). ERMs are therefore attractive targets for anticancer therapies. However, the mechanisms by which ERMs become activated during metastasis remain unknown.

Metastasis is responsible for most cancer deaths (Seyfried & Huysentruyt, 2013). This complex process is driven by the dysregulation of cell motility, which usually plays a crucial role in development and tissue repair. In cancer cells, aberrant signaling pathways activate cytoskeletal rearrangements to promote cell migration and enable invasion and dissemination to distant sites (Fife et al, 2014). In the last decade, evidence showed that cancer

---

[1]Institute for Research in Immunology and Cancer (IRIC), Université de Montréal, Montréal, Canada   [2]Cellular Mechanisms of Morphogenesis During Mitosis and Cell Motility Lab, Montréal, Canada   [3]Molecular Pharmacology Lab, Montréal, Canada   [4]Signaling and Cell Growth Lab, Montréal, Canada   [5]Histology and Molecular Pathology, Montréal, Canada   [6]Université de Montréal, Montréal, Canada   [7]Department of Pathology and Cell Biology, Montréal, Canada   [8]Department of Pharmacology and Physiology, Montréal, Canada   [9]Department of Biochemistry and Molecular Medicine, Montréal, Canada   [10]Rosalind and Morris Goodman Cancer Institute, McGill University, Montreal, Canada

Correspondence: leguay.kevin@gmail.com; michel.bouvier@umontreal.ca; sebastien.carreno@umontreal.ca
*Kévin Leguay and Omaima Naffati contributed equally to this work

cells do not act alone but dialogue with surrounding cells and extracellular matrix components that promote tumor progression and metastasis (Clark & Vignjevic, 2015).

The thromboxane A2 receptor (TBXA2R) is a G protein–coupled receptor (GPCR) that mediates platelet activation and aggregation in response to thromboxane A2 (TXA2), a short half-life (~30 s) prostaglandin derivative (Jones et al, 1985). TXA2 also controls other functions, such as vasoconstriction, inflammation, and angiogenesis (Nakahata, 2008). In a pathological context, TXA2 promotes metastasis of several cancers, including colorectal (Guillem-Llobat et al, 2016), lung (Lucotti et al, 2019), prostate (Nie et al, 2004), and breast (Ekambaram et al, 2011) cancer. In addition, TBXA2R overexpression correlates with poor prognosis in aggressive breast tumor (Watkins et al, 2005), and targeting the TXA2 pathway has been proposed as a strategy against metastasis of these tumors (Li et al, 2017).

Breast cancer is a very heterogeneous disease that can be classified into different subtypes based on the expression levels of different receptors. Among these subtypes, triple-negative breast cancer (TNBC) does not express estrogen, progesterone, and HER2 receptors. Interestingly, several TNBC cells were shown to also overexpress TBXA2R (Orr et al, 2016). Both the high metastatic properties of TNBC cells and the lack of efficient, targeted therapy explain the high mortality rate of this cancer. Although TNBC represents 10–20% of new breast cancer cases, it represents more than 30% of deaths associated with the disease (Bianchini et al, 2022). The molecular mechanisms underlying TNBC metastasis is still poorly understood; thus, identifying the mechanisms that prompt these cells to disseminate could lead to developing more effective treatments for TNBC patients.

We recently demonstrated that RhoA, a small GTPase, activates ERMs at the onset of mitosis (Leguay et al, 2022). Interestingly, several GPCRs, including TBXA2R, are known to engage RhoA to execute some of their functions (Yu & Brown, 2015). This prompted us to investigate the potential functional relationship between TBXA2R, the Rho subfamily of Rho GTPases (RhoA, RhoB, and RhoC), and ERMs. We discovered that TBXA2R engages the heterotrimeric G proteins $G\alpha_{q/11}$ and $G\alpha_{12/13}$ subfamilies, Rho subfamily of Rho GTPases, and their kinase effectors SLK and LOK to activate ERMs in TNBC cells. Furthermore, we showed that activation of TBXA2R in these cells enhances cell migration and invasion in vitro and metastatic colonization in a mouse model of metastasis by a mechanism dependent on the activation of ERMs.

# Results

## Activation of TBXA2R promotes ERM opening and activation

To determine whether TBXA2R activates ERM proteins, we took advantage of ERM conformational enhanced bystander BRET (ebBRET) biosensors that monitor the opening of individual ERMs at the plasma membrane (Leguay et al, 2021). The C terminus of ezrin, radixin, or moesin is fused to the bioluminescent energy donor *Renilla* luciferase (rLucII), whereas their N terminus is anchored at the plasma membrane using a myristoylation signal and polybasic motif. The fluorescent energy acceptor *Renilla* GFP (rGFP) is also targeted to the plasma membrane through its fusion with the prenylated CAAX motif of KRAS (rGFP-CAAX) (Fig S1A). In their closed, inactive conformation, ERMs are globular with a diameter of ~10 nm, whereas they open up to ~40 nm when activated (Liu et al, 2007). Because the transfer of energy between the BRET donor and acceptor is inversely proportional to the distance separating them (Breton et al, 2010), the opening and activation of ERMs result in a decrease in ebBRET signals (Leguay et al, 2021). We first assessed the ability of TBXA2R to promote ERM opening and activation in HEK293T cells, as these cells are well suited to study GPCR signaling using BRET assays (Gales et al, 2005; Breton et al, 2010; Namkung et al, 2016; Namkung et al, 2018; Kobayashi et al, 2019). Because TBXA2R is not expressed in these cells (Atwood et al, 2011), we cotransfected this GPCR with individual ERM biosensors and measured their ebBRET signal. Given the very short half-life of thromboxane A2 that makes its use impractical in many assays, we used the synthetic agonist U46619, a stable, selective, and reliable tool to activate TBXA2R in cellular assays (Coleman et al, 1981). This approach revealed that the three ERM proteins are opened downstream of TBXA2R. Stimulating this GPCR with U46619, decreased ebBRET signals associated with each ERM biosensor (Figs 1A and S1B and C). U46619 promoted ezrin, radixin, or moesin opening with similar low nanomolar potency (Fig 1B), in agreement with the affinity of this agonist for TBXA2R (Kattelman et al, 1986).

The phosphorylation of a conserved threonine residue stabilizes the ERM open, active form. We assessed this last step of ERM activation using an antibody that specifically recognizes phosphorylated ERM (p-ERM) (Carreno et al, 2008). Confirming the results obtained with the ERM biosensors, U46619 promoted a substantial increase in phosphorylation of endogenous ERMs, comparable to the increase triggered by calyculin A, a potent inhibitor of Ser/Thr phosphatases (Fig 1C and D). Furthermore, we found that ERM activation downstream of TBXA2R depends on the catalytic activity of a Ser/Thr kinase because staurosporine, a broad-spectrum kinase inhibitor, prevents ERM phosphorylation upon U46619 treatment (Fig 1C and D). Consistent with these findings, staurosporine also prevented the U46619-induced ebBRET decrease in agreement with the role of phosphorylation in stabilizing the ERM open conformation (Figs 1A and S1B and C). We then characterized the kinetics of ERM activation downstream of TBXA2R and showed that U46619 triggers ERM phosphorylation very rapidly as the levels of p-ERM reach their maximum at 2 min, the earliest time point tested. We also showed that this activation lasts at least 4 h (Figs 1E and F and S1D and E).

Finally, immunofluorescence analysis of p-ERM revealed that TBXA2R activation increases ERM phosphorylation at the cell cortex, where ERMs cross-link the cytoskeleton with the plasma membrane (Fig 1G and H). Altogether, these experiments show that activation of TBXA2R promotes the opening and sustained activation of ERMs.

## TBXA2R engages $G\alpha_{q/11}$, $G\alpha_{12/13}$, the Rho subfamily, and SLK kinases to activate ERMs

Taking advantage of the versatility of HEK293 cells for transfection and loss-of-function experiments (Pulix et al, 2021), we

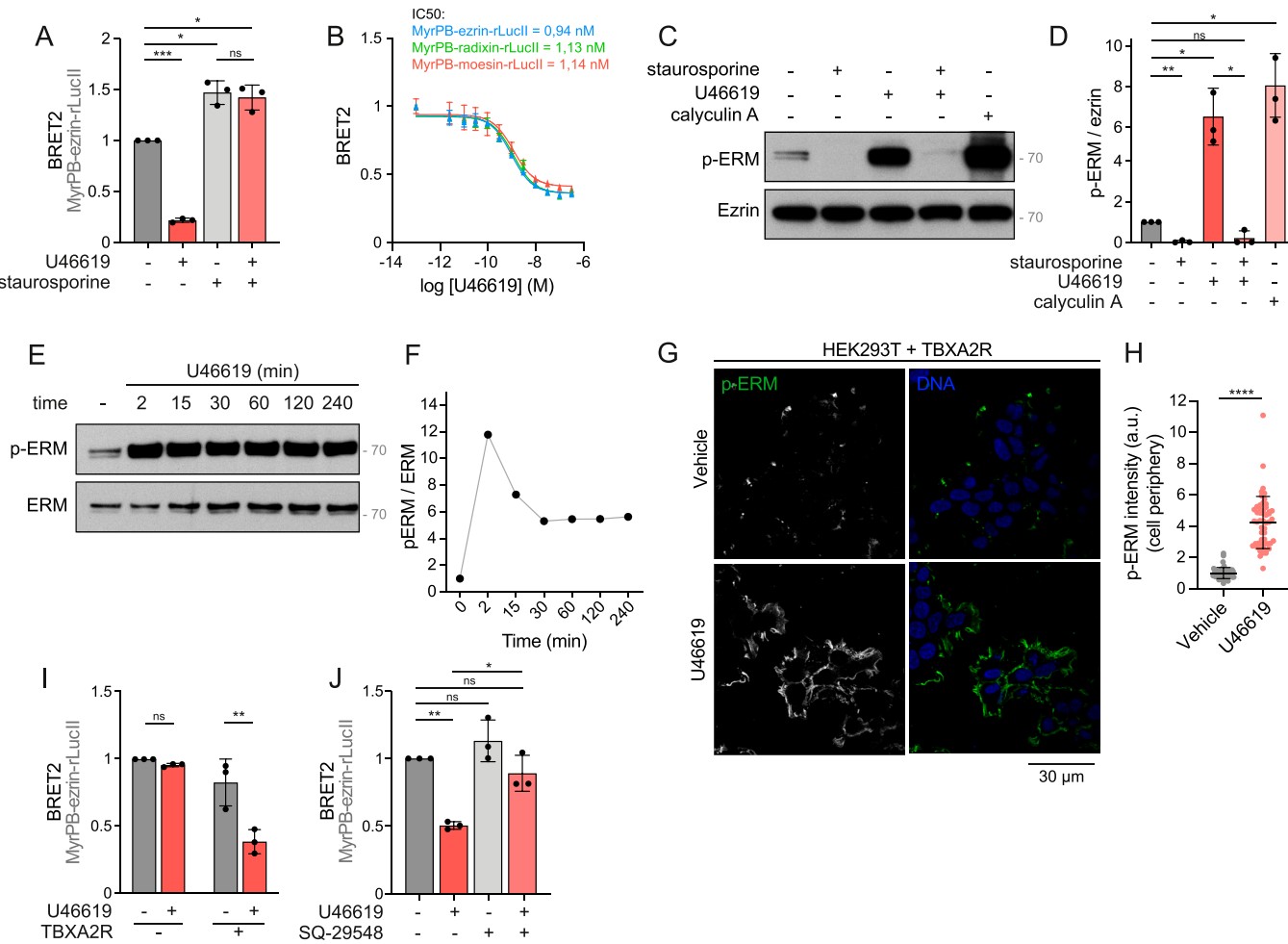

**Figure 1. TBXA2R stimulation triggers ERM activation in HEK293T cells.**
**(A)** ebBRET signals were measured in HEK293T cells expressing MyrPB-ezrin-rLucII, cotransfected with TBXA2R (HEK293T-TBXA2R), and treated with vehicle, 100 nM U46619 for 5 min, and/or 100 nM staurosporine for 30 min. **(B)** ebBRET signals measured in HEK293T-TBXA2R cells expressing MyrPB-E,R,M-rLucII and treated with increasing concentrations of U46619 for 5 min. **(C, D)** Immunoblot of HEK293T-TBXA2R cells treated with vehicle, 100 nM staurosporine for 30 min, 100 nM U46619 for 5 min, and/or 100 nM calyculin A for 10 min (C). **(D)** p-ERM over ezrin signals was quantified and normalized to the vehicle (D). **(E, F)** Immunoblot of HEK293T-TBXA2R cells treated with 100 nM U46619 for the indicated times (E). **(F)** p-ERM over ERM signals was quantified and normalized to the vehicle (F). **(G, H)** Immunofluorescence of HEK293T-TBXA2R cells treated with vehicle or 100 nM U46619 for 5 min (G). **(H)** p-ERM staining at the cell periphery was quantified and normalized to cells treated with vehicle (H). **(I)** ebBRET signals measured in HEK293T cells expressing MyrPB-ezrin-rLucII and cotransfected with or without TBXA2R and treated with vehicle or 100 nM U46619 for 5 min. **(J)** ebBRET signals were measured in HEK293T-TBXA2R cells expressing a MyrPB-ezrin-rLucII biosensor and treated with vehicle, 100 nM U46619 for 5 min, and/or 1 μM SQ-29548 for 30 min. ebBRET signals **(A, B, I, J)** represent the mean ± s.d. of three independent experiments. **(G)** Immunoblots (C, E) and immunofluorescences (G) are representative of three independent experiments. P-ERM quantifications (D, F, H) represent the mean ± s.d. of three independent experiments. **(A, B, D, F, H, I, J)** Dots represent independent experiments (A, D, I, J), individual cells (H), or the mean of independent experiments (B, F). **(A, D, I, J, H)** *P*-values were calculated using a one-sample *t* test (J), two-tailed unpaired *t* test (H), or two-tailed paired *t* test (A, D, I) except for comparisons made with the normalizing condition (vehicle) where one-sample *t* test was applied. \**P* < 0.05, \*\**P* < 0.01, \*\*\**P* < 0.001, \*\*\*\**P* < 0.0001, ns, not significant.

aimed to identify the signaling pathway that TBXA2R engages in activating ERMs. Because TBXA2R activates ezrin, radixin, and moesin similarly (Fig 1B), we focused on the ezrin biosensor as a proxy to study ERM regulation. In accordance with U46619-activating ERMs through TBXA2R, U46619 did not affect ebBRET signals associated with the ezrin biosensor in cells that were not transfected with TBXA2R (Fig 1I). Further demonstrating the specificity of this newly discovered signaling axis, SQ-29548, a highly selective TBXA2R antagonist (Ogletree et al, 1985), prevented ezrin opening upon TBXA2R activation with U46619 (Fig 1J).

When activated, TBXA2R engages $G\alpha_{q/11}$ and $G\alpha_{12/13}$ subfamily members that relay the signal to different effectors (Nakahata, 2008; Avet et al, 2022). To characterize the role of each of these two $G\alpha$ subfamilies in ERM activation, we used CRISPR-engineered HEK293 cells lacking the $G\alpha_{q/11}$ (Schrage et al, 2015) or $G\alpha_{12/13}$ (Devost et al, 2017) subunits ($\Delta G\alpha_{q/11}$ and $\Delta G\alpha_{12/13}$ cells, respectively). In $\Delta G\alpha q/11$ cells and $\Delta G\alpha 12/13$, TBXA2R activation still promoted ezrin opening (Fig 2A). We then tested whether $G\alpha_{q/11}$ and $G\alpha_{12/13}$ subfamilies exert redundant functions in ERM activation downstream of TBXA2R. Treatment of $\Delta G\alpha_{12/13}$ cells with YM-254890, a selective $G\alpha_{q/11}$ subunit inhibitor (Takasaki et al, 2004),

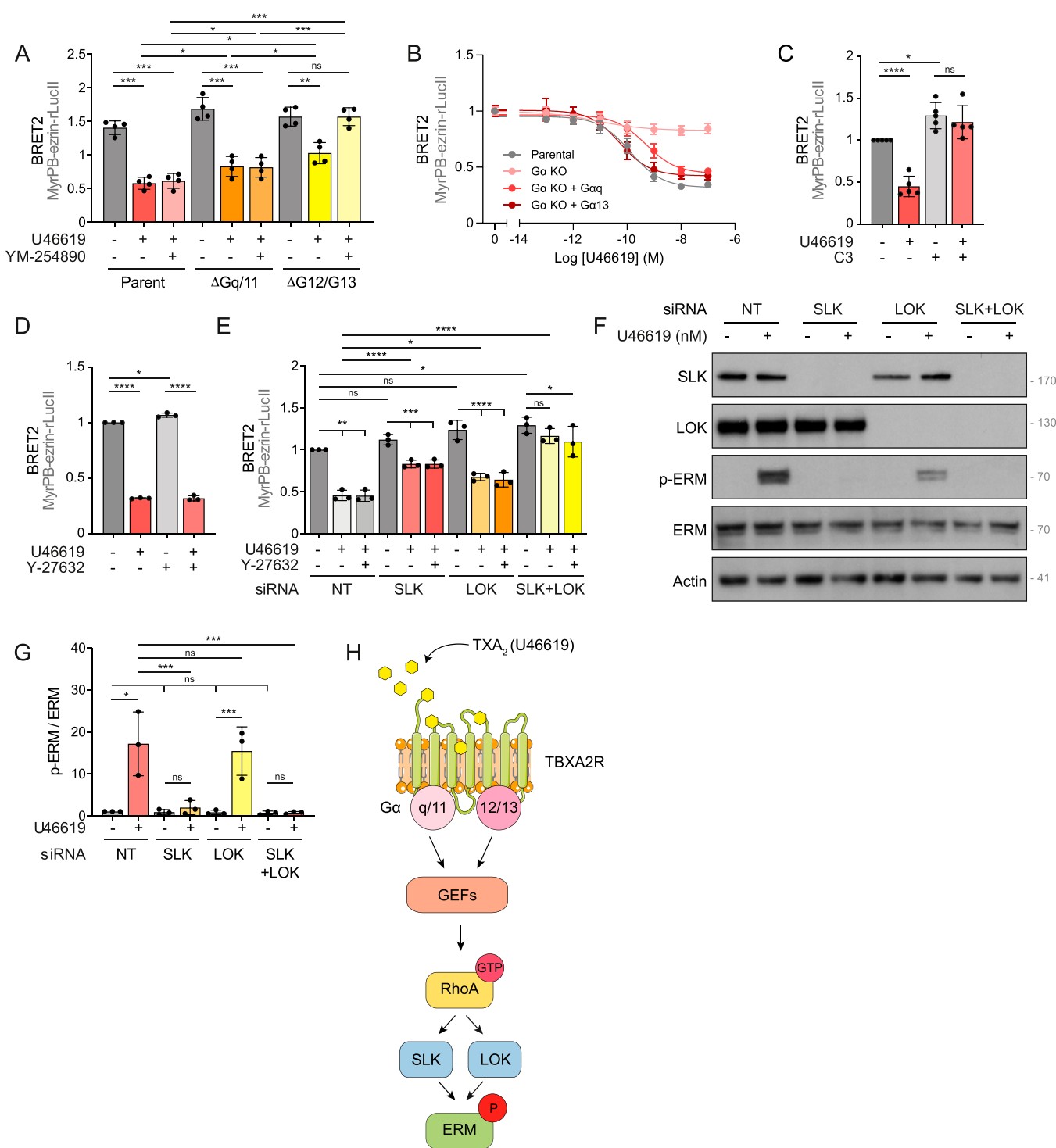

**Figure 2.   TBXA2R activates ERMs through Rho GTPases of the Rho subfamily and its kinase effectors SLK/LOK.**
**(A)** ebBRET signals measured in parental HEK293T cells or HEK293T cells knocked out for Gq/11 (ΔGq/11) or G12/13 (ΔG12/13), co-expressing TBXA2R and MyrPB-ezrin-rLucII biosensor, and treated with 100 nM U46619 for 5 min and/or 1 μM YM-254890 for 30 min. **(B)** ebBRET signals measured in parental HEK293T cells, HEK293T cells knocked out for all Gα subunits, or HEK293T cells knocked out for all Gα subunits and overexpressing Gαq or Gα13. These cell lines co-expressed TBXA2R and MyrPB-ezrin-rLucII biosensor and were treated with a dose–response of U46619 for 5 min. **(C, D)** ebBRET signals measured in HEK293T-TBXA2R expressing MyrPB-ezrin-rLucII biosensor and treated with 100 nM U46619 for 5 min and/or 1 μg/ml C3 transferase for 6 h (C) or 10 μM Y-27632 for 4 h (D). **(E)** ebBRET signals measured in HEK293T-TBXA2R expressing MyrPB-ezrin-rLucII biosensor, transiently transfected with nontarget siRNA (NT) or siRNA targeting SLK and/or LOK, and treated with 100 nM U46619 for 5 min and/or 10 μM Y-27632 for 4 h. **(F, G)** Immunoblot of HEK293T-TBXA2R cells transiently transfected with nontarget siRNA (NT) or siRNA targeting SLK and/or LOK and treated with 100 nM U46619 for 5 min (F). **(G)** p-ERM over ERM signals was quantified and normalized to NT incubated with vehicle (G). **(H)** Proposed model for the signaling pathway downstream of TBXA2R that activates ERMs in HEK293T-TBXA2R cells. ebBRET signals (A, B, C, D, E) represent the mean ± s.d. of at least three independent experiments. Immunoblot (F) is representative of three independent experiments. P-ERM quantifications (G) represent the mean ± s.d. of three

completely inhibited ezrin opening upon TBXA2R activation (Fig 2A). Consistently, YM-254890 had no effect on ezrin opening in control parental cells, in line with the observation that deletion of Gαq/11 alone does not impair ERM activation. Then, we reasoned that if either Gαq/11 or Gα12/13 alone is sufficient to support ERM activation downstream of TBXA2R, reintroducing a single Gα subunit into a Gα-null background should restore signaling. To test this, we used a HEK293 Gα-null cell line lacking all Gα subunits (Gα KO) (Ono et al, 2023) and reexpressed either Gαq or Gα13 individually. Reintroduction of either subunit was sufficient to restore ezrin opening upon TBXA2R activation (Fig 2B), demonstrating that Gαq/11 and Gα12/13 act redundantly to mediate ERM activation downstream of TBXA2R.

Among the effectors of $G\alpha_{q/11}$ and $G\alpha_{12/13}$, Rho GTPases of the Rho subfamily were shown to promote phosphorylation and activation of ERMs (Shaw et al, 1998; Bagci et al, 2020; Leguay et al, 2022). Indeed, treatment with the exoenzyme C3 transferase, a selective inhibitor of these small GTPases (Wilde & Aktories, 2001), prevented ezrin opening upon TBXA2R stimulation (Fig 2C), establishing that the Rho subfamily acts downstream of $G\alpha_{q/11}$ and $G\alpha_{12/13}$ subunits to activate ERMs.

We next aimed to identify which kinase(s) act(s) downstream of TBXA2R to activate ERMs. GTPases of the Rho subfamily were shown to directly bind and activate two families of Ser/Thr kinases that can phosphorylate ERMs: the Rho-associated protein kinases ROCK1 and ROCK2 (Matsui et al, 1996) and the Ste20-like kinase (SLK) and lymphocyte-oriented kinase (LOK) paralogs (Viswanatha et al, 2012; Bagci et al, 2020; Leguay et al, 2022). We previously showed that treating HEK293T cells with 10 μM Y-27632 for 4 h effectively inhibits ROCK kinases, as evidenced by reduced phosphorylation of myosin light chain II, a well-characterized ROCK substrate (Leguay et al, 2022). Using the same conditions, we now show that ROCK1 and ROCK2 activity is not essential for ERM activation downstream of TBXA2R because Y-27632 did not affect ezrin opening after U46619 stimulation (Fig 2D). In contrast, transient depletion of SLK by RNAi reduced ezrin opening downstream of TBXA2R activation (Fig 2E). Although the depletion of LOK led to a slight inhibition of ezrin opening, the codepletion of both LOK and SLK paralogs completely prevented ezrin opening downstream of TBXA2R (Fig 2E). Consistent with kinases of the SLK family being the main kinases that relay the signal between TBXA2R and ERMs, inhibition of ROCK kinases with Y-27632 did not further reduce ezrin opening upon SLK, LOK, or SLK/LOK transient depletion (Fig 2E). We then observed that LOK depletion led to a partial reduction in ERM phosphorylation downstream of TBXA2R activation (Fig 2F and G). SLK depletion, however, almost completely abolished ERM phosphorylation, identifying SLK as the principal kinase mediating TBXA2R-dependent ERM activation in these cells. Codepletion of LOK and SLK fully prevented detectable ERM phosphorylation,

indicating that these homologous kinases cooperate to drive ERM activation.

Altogether, these results establish that TBXA2R activates GTPases of the Rho subfamily through both $G\alpha_{q/11}$ and $G\alpha_{12/13}$, leading to ERM activation via kinases of the SLK/LOK family (Fig 2H).

## TBXA2R activates ERMs in TNBC cells

Having shown that TBXA2R activates ERMs, we hypothesized that this signaling axis could play a role in the biology of cancer cells. Given that TBXA2R and ERMs have previously been shown to regulate motility and invasion of TNBC cells in separate studies (Orr et al, 2016; Qin et al, 2020), we investigated the potential involvement of the TBXA2R-ERM signaling axis in human TNBC. Upon interrogation of transcriptomic data from TNBC samples obtained from the Cancer Genome Atlas of Invasive Breast Carcinoma (Cancer Genome Atlas Network, 2012), we observed that TNBC exhibited higher levels of TBXA2R mRNA expression compared with normal breast tissues (Fig 3A). Further analysis revealed that only a subset of TNBC patients exhibited higher mRNA expression levels of this GPCR compared with normal breast tissues (Fig 3B). Interestingly, our analysis showed that TBXA2R high mRNA expression is associated with poorer overall survival. Such association was not observed in non-TNBC subtypes (Fig 3C and D). To investigate whether ERMs are activated downstream of TBXA2R signaling in cancer tissues from TNBC patients, we examined the levels of phosphorylated ERMs in TNBC biopsies using tissue microarrays (TMAs). Our analysis revealed a positive correlation between TBXA2R protein expression and levels of activated p-ERM in 55 TNBC samples (Fig 3E and F, r = 0.50, P < 0.0001). Based on the levels of p-ERM and TBXA2R protein expression, we identified three distinct subgroups of TNBC. The first subgroup exhibited low levels of both p-ERMs and TBXA2R, suggesting that ERM activation or TBXA2R plays a minor role, if any, in TNBC biology for this subgroup. The second showed high levels of p-ERMs but low TBXA2R expression, indicating possible alternative pathways for ERM activation. The third demonstrated high levels of both p-ERMs and TBXA2R, indicating that TBXA2R may activate ERMs in this subgroup. Notably, none of the TNBC samples exhibited high TBXA2R expression alongside low p-ERM levels. Although the sample size is limited, this is consistent with a functional link between TBXA2R expression and ERM activation. Finally, we investigated whether TBXA2R promotes the phosphorylation of ERMs across a panel of different TNBC cell lines. In the absence of an antibody that reliably detects endogenous TBXA2R in cultured cells by Western blot or immunofluorescence, we selected six TNBC cell lines based on TBXA2R mRNA expression levels reported in the Cancer Dependency Map Portal (Tsherniak et al, 2017) (Fig S2A). Interestingly, higher expression levels of TBXA2R mRNA were associated with a greater ability of the U46619 agonist to promote ERM phosphorylation (Fig 3G and H). Consistent with the expected role of ERMs at the cell cortex, activation

---

independent experiments. **(A, B, C, D, E, F)** Dots represent independent experiments (A, C, D, E, F) or the mean of independent experiments (B). **(A, C, D, E, G)** P-values were calculated using Holm–Sidak's multiple comparisons test with a single pooled variance (A, E, G) or using two-tailed paired t test (C, D) except for comparisons made with the normalizing condition (vehicle, (C, D); NT + vehicle, (E, G)) where a one-sample t test was applied. *P < 0.05, **P < 0.01, ***P < 0.001, ****P < 0.0001, ns, not significant.

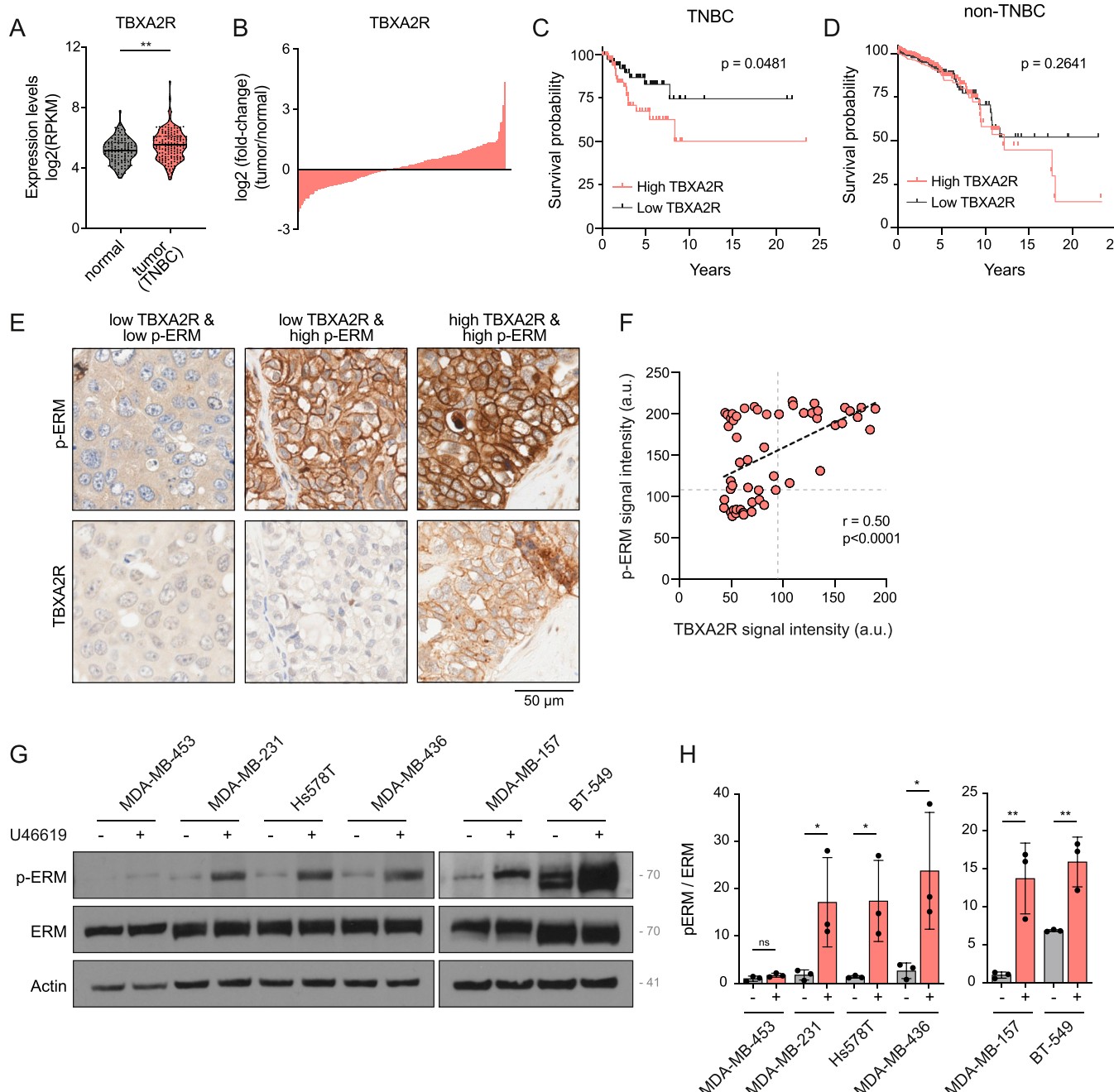

**Figure 3. TBXA2R activates ERMs in triple-negative breast cancer (TNBC) cells.**

**(A, B, C, D)** Analysis of TBXA2R mRNA expression in TNBC tissues and associated overall survival. Data were extracted from The Cancer Genome Atlas (TCGA). **(A)** TNBC cells present higher TBXA2R mRNA expression when compared to the normal breast tissue. **(B)** Within TNBC cases, a subset of tumors shows elevated TBXA2R mRNA expression relative to the normal breast tissue. **(C, D)** Kaplan–Meier curves show that high expression of TBXA2R is associated with shorter overall survival in TNBC patients but not in non-TNBC patients (D). **(E, F)** Representative IHC staining of p-ERM and TBXA2R in TNBC tissue (E). **(F)** p-ERM and TBXA2R signal intensities were quantified at the cell cortex (n = 55, (F)). Linear regression and Pearson's correlation score are represented on the graph. **(G, H)** Immunoblot of 6 TNBC cell lines treated with vehicle or 100 nM U46619 for 5 min (G). **(H)** p-ERM over ERM signals was quantified and normalized to MDA-MB-453 treated with vehicle (left) or MDA-MB-157 treated with vehicle (right) (H). Immunoblot (G) is representative of three independent experiments. P-ERM quantification (H) represents the mean ± s.d. of three independent experiments. **(A, F, H)** Dots represent independent samples (A, F) or individual experiment (H). **(A, C, D, F, H)** P-values were calculated using a two-tailed unpaired *t* test (A), log-rank (Mantel–Cox) test (C, D), Pearson's correlation (F), or two-tailed paired *t* test (H). *P < 0.05, **P < 0.01, ns, not significant.

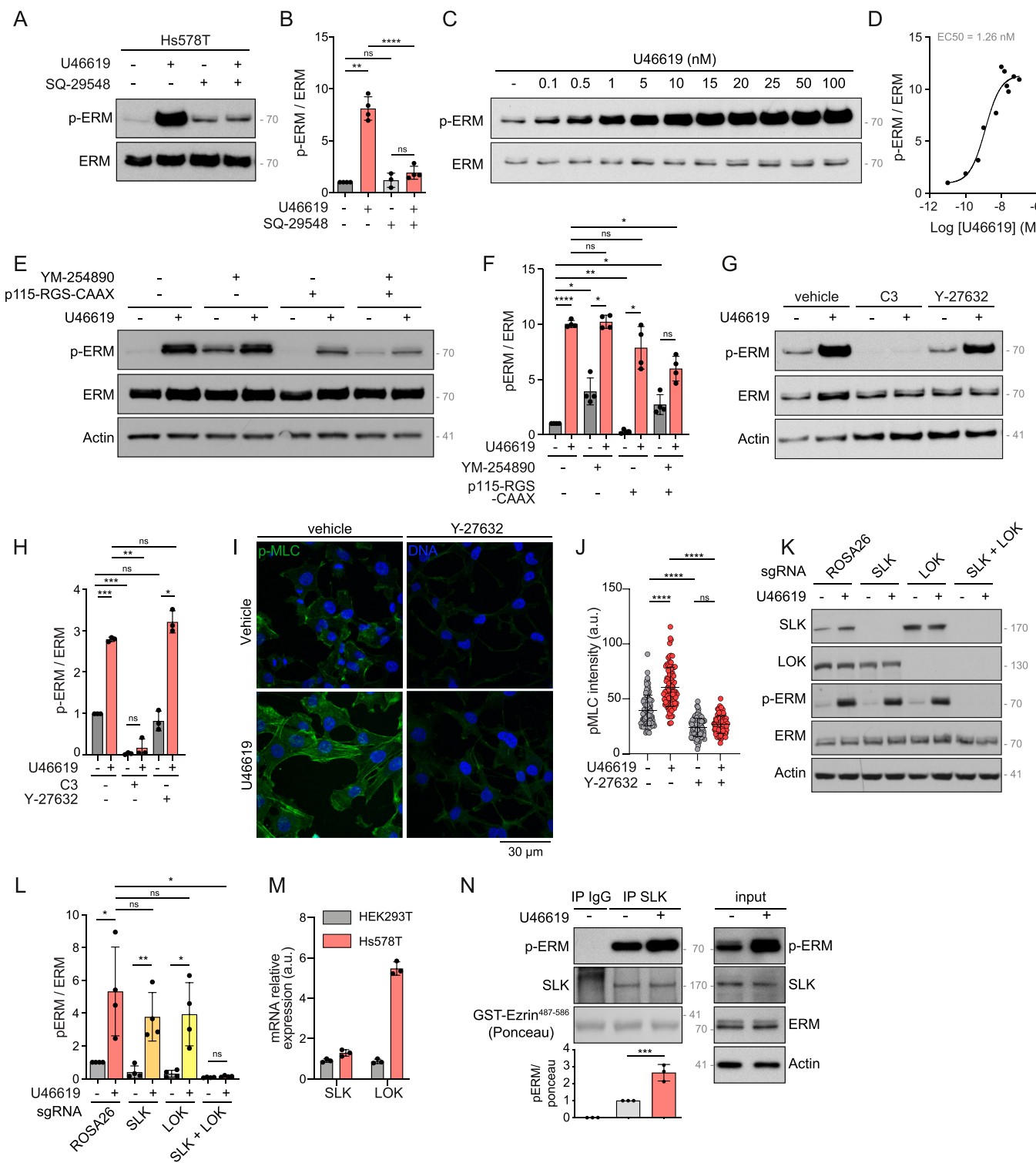

**Figure 4. TBXA2R activates ERMs through Gα$_{q/11}$ and Gα$_{12/13}$, Rho GTPases of the Rho subfamily, and SLK/LOK in Hs578T.**
**(A, B)** Immunoblot of Hs578T cells that were treated with vehicle, 100 nM U46619 for 5 min, and/or 1 μM SQ-29548 for 30 min (A). **(B)** P-ERM over ERM signals was quantified and normalized to the vehicle (B). **(C, D)** Immunoblot of Hs578T cells treated with U46619 for 5 min with the indicated concentrations (C). **(D)** P-ERM over ERM signals was quantified and normalized to the vehicle (D). **(E, F, G, H)** Immunoblots of Hs578T cells treated with vehicle or 100 nM U46619 for 5 min and 1 μM YM-254890 for 30 min, and/or transfected with 1 μg p115-RGS-CAAX (E), and treated with vehicle or 100 nM U46619 for 5 min, 1 μg/ml C3 transferase for 6 h, and/or 10 μM Y-27632 for 4 h (G). **(F, H)** P-ERM over ERM signals was quantified and normalized to the vehicle (F, H). **(I, J)** Immunofluorescence of Hs578T pretreated with vehicle or 10 μM Y-27632 for 4 h and cotreated with 100 nM U46619 for 5 min before cell fixation (I). **(J)** Total pMLC intensity was quantified (J). **(K, L)** Immunoblot of CRISPR-Cas9–mediated SLK and/or LOK Hs578T knockout cells treated with vehicle or 100 nM U46619 for 5 min (K). **(L)** P-ERM over ERM signals was quantified and normalized to control cells (ROSA26) treated with vehicle (L). **(M)** Relative mRNA expression levels of SLK and LOK in HEK293T and Hs578T measured by RT–qPCR. **(N)** Immune complex kinase assay of

of endogenous TBXA2R by its agonist increased ERM phosphorylation at the plasma membrane of all examined TNBC cell lines (Fig S2B–M).

## TBXA2R activates ERMs through $G_{q/11}$ and $G_{12/13}$, GTPases of the Rho subfamily, and SLK/LOK in Hs578T TNBC cells

To further investigate the significance of the TBXA2R-ERM signaling axis in TNBC, we focused on the Hs578T cell line, which showed a median level of TBXA2R mRNA expression among the six TNBC cell lines tested (Fig S2A). Hs578T cells are well characterized and have been extensively used in cancer research, providing a reliable model for studying the molecular mechanisms underlying cell motility, invasion, and metastasis (Koedoot et al, 2019; Yankaskas et al, 2019). We first confirmed the pharmacological selectivity of the TBXA2R agonist in Hs578T cells by showing that the TBXA2R antagonist SQ29548 blocks U46619-promoted ERM phosphorylation downstream of TBXA2R (Fig 4A and B). As we observed in HEK293T cells, U46619 activated ERMs in Hs578T cells at nanomolar concentrations (Figs 1B and 4C and D). We tested the role of the two Gα subfamilies using YM-254860 to inhibit $G\alpha_{q/11}$ subunits and by overexpressing p115-RGS-CAAX, a previously characterized dominant-negative construct that inhibits $G\alpha_{12/13}$ activity (Lukasheva et al, 2020). In this construct, the RGS domain of p115-RhoGEF that inactivates $G\alpha_{12/13}$ signaling (p115-RGS) is targeted at the plasma membrane using a CAAX motif. As observed in HEK293T cells, inhibition of $G\alpha_{q/11}$ alone, using YM254890, did not reduce the activation of ERMs downstream of TBXA2R. In contrast, inhibition of $G\alpha_{12/13}$ partially prevented ERM activation as measured by Western blot analysis of p-ERM (Fig 4E and F). Co-inhibition of both Gα subfamilies further reduced ERM activation (Fig 4E and F). In addition, inhibition of GTPases of the Rho subfamily using exoenzyme C3 transferase, but not inhibition of ROCK kinases using Y-27632, prevented phosphorylation of ERMs upon TBXA2R activation (Fig 4G and H). To confirm ROCK inhibition, we monitored phospho-myosin II levels by immunofluorescence. U46619 increased phospho-myosin II staining compared with untreated cells, whereas Y-27632 profoundly reduced this signal in both control and U46619-treated cells (Fig 4I and J). These results confirm effective ROCK inhibition and indicate that ROCK kinases are dispensable for ERM activation downstream of TBXA2R in Hs578T cells. Finally, although SLK depletion was sufficient to markedly reduce ERM phosphorylation in HEK293T cells, single CRISPR-Cas9 knockout of either SLK or LOK in Hs578T cells did not reduce ERM phosphorylation downstream of TBXA2R activation (Fig 4K and L). Yet, the double knockout of these two paralogs abolished ERM activation upon TBXA2R stimulation. This requirement for dual kinase knockout in Hs578T cells may reflect differences in kinase expression. As mRNA quantification (Fig 4M) demonstrated, LOK is expressed at much higher levels in

Hs578T cells than in HEK293T cells, whereas SLK expression is comparable between the two cell lines.

Confirming the role of these kinases in the TBXA2R-GTPases of the Rho subfamily signaling pathway, SLK kinase activity increased when the GPCR was stimulated with U46619 (Fig 4N). Taken together, these data establish that $G\alpha_{q/11}$ and $G\alpha_{12/13}$ subunits, GTPases of the Rho subfamily, and SLK and LOK act downstream of TBXA2R to activate ERMs in Hs578T cells.

## TBXA2R activates SLK/LOK and moesin to potentiate Hs578T cell 2D motility, 3D invasion, and metastatic colonization

We then investigated the role of the TBXA2R-ERM signaling axis in TNBC cell motility and invasion in vitro. To this end, we deleted individual ERMs by CRISPR-Cas9 editing of Hs578T cells. Although the deletion of ezrin or radixin did not affect the phosphorylation of endogenous ERMs promoted by TBXA2R activation, the deletion of moesin almost totally abolished ERM phosphorylation (Fig 5A and B). We confirmed this finding by stably knocking down ezrin, radixin, or moesin in Hs578T cells with shRNAs targeting each ERM (Fig S3A). Because TBXA2R activates ezrin, radixin, or moesin similarly in an engineered system (Fig 1B), this suggests that moesin is the most abundant ERM expressed in Hs578T cells. This hypothesis was confirmed by analyzing gene expression in Hs578T cells. Single-cell RNA-sequencing data available in the Single Cell Expression Atlas (Papatheodorou et al, 2020) reveal that moesin mRNA is expressed at the highest level among the three ERMs in Hs578T cells (Fig S3B). We thus decided to further study the role of the TBXA2R-ERM signaling axis using the Δmoesin Hs578T cells.

To measure the motility of Hs578T cells, we used a modified version of the wound-healing scratch assay (Oris) (Fig 5C). Cells migrating into the free area were followed for 6 h, a timescale during which ERMs remained activated by U46619 in Hs578T cells (Fig S3C and D). We found that upon TBXA2R stimulation, Hs578T cells migrated ~1.5 faster than nonstimulated cells (Fig 5D and E). The deletion of moesin by CRISPR-Cas9 using two independent single guide RNA (Fig S3E and F) significantly slowed down the spontaneous cell migration compared with controls (Fig 5D), confirming the importance of ERMs for cell migration (Arpin et al, 2011). Furthermore, we found that the TBXA2R-promoted increase of Hs578T cell migration is an ERM-dependent process, as demonstrated by the lack of increased motility in the two independent Δmoesin Hs578T cell lines after activation of this GPCR (Fig 5D). This finding underscores the importance of moesin in mediating the effect of TBXA2R on Hs578T cell migration. Confirming this, the depletion of moesin by shRNA resulted in the same cell motility defects downstream of TBXA2R activation as the one observed in the Δmoesin cells

---

endogenous SLK immunoprecipitated from Hs578T cells incubated with vehicle or 100 nM U46619 for 5 min. The total lysate (input) is shown in the right panel. p-ERM signals over Ponceau staining were quantified and normalized to immunoprecipitated SLK treated with vehicle (lane 2). Immunoblots (A, C, E, K, N) are representative of at least two independent experiments. P-ERM quantifications (B, D, F, H, L) represent the mean ± s.d. of at least two independent experiments. pMLC quantifications (J) represent the mean ± s.d. of three independent experiments. **(B, D, F, H, J, L, M)** Dots represent independent experiments (B, F, H, L, M), individual cells (J), or the mean of independent experiments (D). **(B, F, H, J, L, N)** *P*-values were calculated using Holm–Sidak's multiple comparisons test with a single pooled variance (B, F, H, J, L) or a one-sample *t* test (N) except for comparisons made with the normalizing condition (vehicle, B, F, H; sgROSA26 + vehicle, L) where one-sample *t* test was applied. \**P* < 0.05, \*\**P* < 0.01, \*\*\**P* < 0.001, \*\*\*\**P* < 0.0001, ns, not significant.

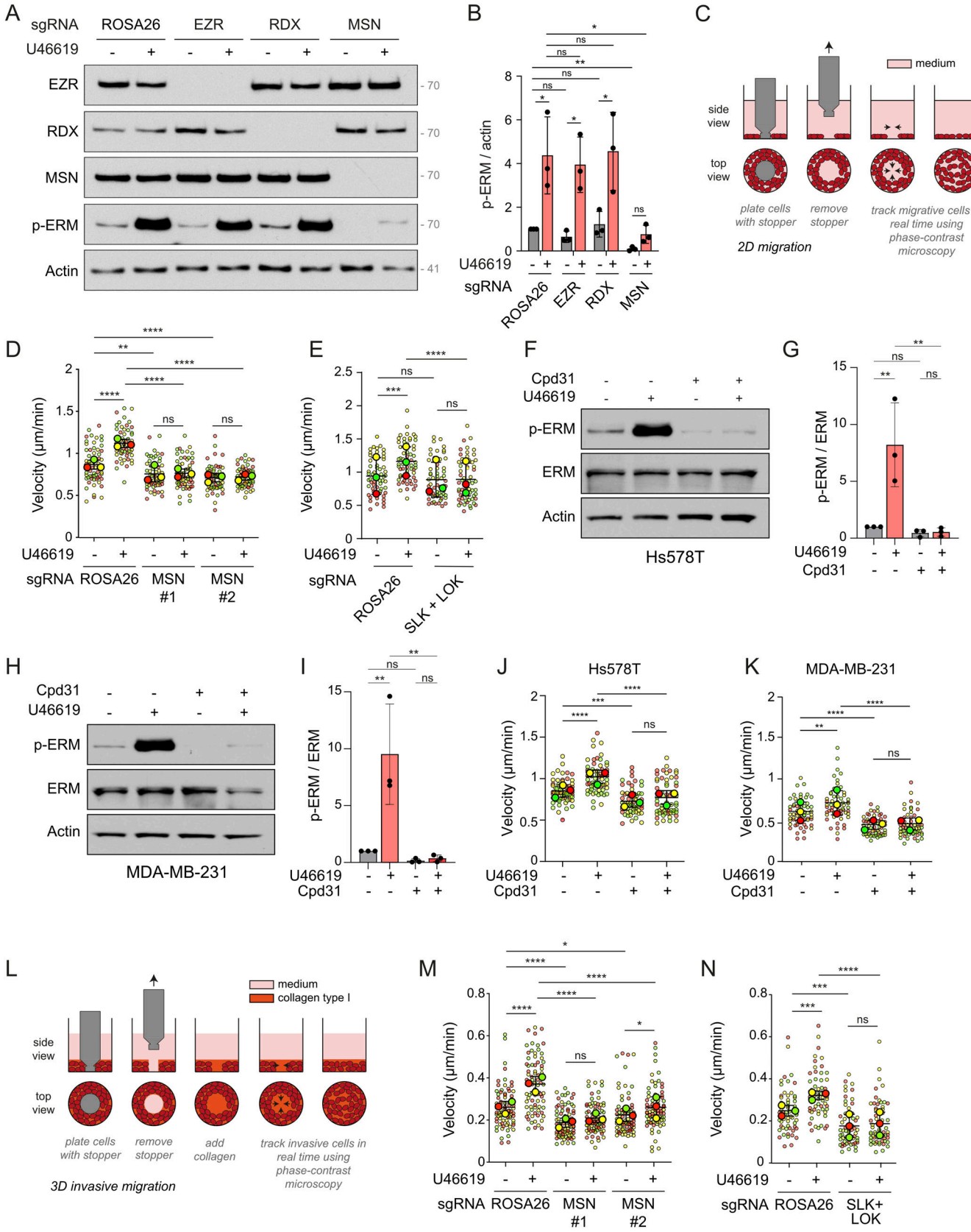

(Fig S3G). We also found that SLK and LOK double knockout completely abrogated the TBXA2R-promoted migration confirming that they are the kinases activating ERMs downstream of TBXA2R and demonstrating their essential role in the migration resulting from activation of this GPCR (Fig 5E).

To further validate the involvement of SLK/LOK kinases and assess the generality of our findings, we examined a second TNBC cell line, MDA-MB-231, which expresses comparable levels of TBXA2R (Fig S2A). Treatment of MDA-MB-231 cells with Cpd31, a recently identified selective inhibitor of SLK/LOK (Serafim et al, 2021; Marshall-Burghardt et al, 2024), abolished ERM phosphorylation downstream of TBXA2R activation, indicating that these kinases are also required for ERM activation in this other TNBC cell line (Fig 5H and I). In parallel, Cpd31 prevented ERM phosphorylation in U46619-treated Hs578T cells, providing pharmacological confirmation of the genetic evidence obtained using SLK/LOK knockouts (Figs 4K and L and 5F and G). In both Hs578T and MDA-MB-231 cells, Cpd31 significantly reduced migration speed, with or without TBXA2R activation (Fig 5J and K), further supporting the essential role of SLK/LOK in TBXA2R-driven cell motility.

To investigate the role of the TBXA2R-ERM signaling axis during cell invasion, Hs578T cells were embedded in an extracellular collagen matrix (ECM) using the wound-healing scratch assay (Oris) (Fig 5L). Cells invading the free zone filled with ECM, using serum as a chemoattractant, were then tracked. This revealed that stimulation of TBXA2R with U46619 increases the speed of Hs578T cell invasion into the ECM by ~1.5-fold (Fig 5M and N). As observed for 2D cell migration, moesin and SLK/LOK were necessary for TBXA2R to potentiate cell invasion into the ECM. Indeed, moesin or SLK/LOK knockouts prevented TBXA2R from potentiating invasion of TNBC cells (Fig 5M and N). Our findings establish that activation of ERMs via SLK/LOK downstream of TBXA2R is a key mechanism by which this GPCR promotes cell migration and invasion in TNBC cells.

Finally, we assessed whether TBXA2R signaling could potentiate TNBC metastasis through ERM activation in vivo. We performed a tail vein metastasis assay in immunodeficient NSG mice treated or not with the TBXA2R agonist U46619 (Fig 6A). To mimic a pathological context in which TXA2, the natural ligand of TBXA2R, is chronically elevated, we continuously administered this stable agonist. Such sustained TXA2 production has been reported in several cancers (Matsui et al, 2012), and can arise from activated platelets, endothelial cells, macrophages, neutrophils, or tumor cells within the tumor microenvironment and metastasis sites

(Ashton et al, 2022). U46619 was well tolerated, with no changes in body weight or liver weight in control mice (Fig S4A and B).

Twenty-four days after injection of control (ROSA26) or Δmoesin Hs578T cells, we examined metastases in the lungs and liver after tail vein injection, two common sites of TNBC dissemination. In the lungs, macroscopic analysis revealed that U46619 treatment increased tumor burden compared with untreated controls (Fig 6B). Mice injected with Δmoesin Hs578T cells showed fewer lung metastases than those injected with control cells, indicating that moesin contributes to lung dissemination. Moreover, U46619 had no detectable effect in the absence of moesin, demonstrating that TBXA2R-driven lung metastases are moesin-dependent. However, the extensive metastatic spread in the lungs of U46619-treated mice at this time point precluded reliable histological quantification by IHC.

We next examined the liver histopathological analysis of serial sections. In mice injected with control Hs578T cells, this revealed that U46619 increased both metastatic nodule size and overall liver tumor burden, without affecting the number of nodules per animal (Figs 6C–E and S4C). TBXA2R activation also increased the proportion of mice with a metastatic liver burden exceeding 10% of the surface area, from 25% (2/8 mice) to 60% (6/10 mice) (Fig 6E). In contrast, mice injected with Δmoesin Hs578T cells presented fewer metastatic nodules than control cell–injected mice, confirming that moesin contributes to metastasis. Furthermore, U46619 treatment did not increase either nodule size or liver tumor load in the absence of moesin (Figs 6C–E and S4C) confirming the important role of this ERM downstream of the receptor for metastasis. None of the Δmoesin-injected mice exhibited a metastatic burden above 10% of the liver surface (Fig 6E). These differences are most likely not attributable to altered tumor growth, as U46619 did not affect the proliferation of control or Δmoesin cells in vitro (Fig S4D), and Ki-67 indices in liver metastases were comparable across all groups (Fig 6F and G).

Together, these observations demonstrate that moesin is required for this TNBC cell line to colonize the lung and liver, and that TBXA2R activation enhances metastatic progression in a moesin-dependent manner.

## Discussion

Our study sheds light on the involvement of a GPCR-ERM signaling pathway in cancer cell biology. We have discovered that the TBXA2R

---

**Figure 5. Moesin mediates TBXA2R-induced motility and invasion of Hs578T cells in vitro.**
**(A, B)** Immunoblot of Hs578T cells after CRISPR-Cas9–mediated knockout of ROSA26 (control), ezrin (EZR), radixin (RDX), and moesin (MSN) and treated with 100 nM U46619 for 5 min (A). **(B)** p-ERM over actin signals was quantified and normalized to ROSA26 treated with vehicle (B). **(C)** Schematic representation of in vitro 2D migration experimental protocol. **(D, E)** Quantification of cell velocity during 2D migration of Hs578T cells after CRISPR-Cas9–mediated knockout of ROSA26 (control) or MSN using two independent sgRNAs (D), and double knockout of SLK and LOK (E), treated with vehicle or 10 nM U46619 for 6 h. **(F, G, H, I)** Immunoblots of Hs578T (F) or MDA-MB-231 (H) cells cotreated with 10 μM Cpd31 and/or 100 nM U46619. **(G, I)** p-ERM over ERM signals was quantified and normalized to vehicle (G, I). **(J, K)** Quantification of cell velocity during 2D migration of Hs578T (J) or MDA-MB-231 (K) cells cotreated with 10 μM Cpd31 and/or 5 or 20 nM U46619. **(L)** Schematic representation of in vitro 3D invasive migration experimental protocol. **(M, N)** Quantifications of cell velocity during 3D invasive cell migration of Hs578T cells after CRISPR-Cas9–mediated knockout of ROSA26 (control) or MSN using two independent sgRNAs (M), and double knockout of SLK and LOK (N), treated with vehicle or 10 nM U46619 for 24 h. Immunoblots (A, F, H) are representative of three independent experiments. P-ERM quantifications (B, G, I) represent the mean ± s.d. of three independent experiments. Velocity quantifications (D, E, J, K, M, N) are presented as superplots. The speed of individual cells is shown as small dots, color-coded by independent experiment. The mean ± s.d. of each biological replicate (n = 3 independent experiments) is shown as larger dots. **(B, D, E, G, I, J, K, M, N)** $P$-values were calculated using Holm–Sidak's multiple comparisons test with a single pooled variance (B, D, E, G, J, K, M, N) except for comparisons made with the normalizing condition (ROSA26 + vehicle, (B); vehicle, (G, I)) where a one-sample $t$ test was applied. $*P < 0.05$, $**P < 0.01$, $***P < 0.001$, $****P < 0.0001$, ns, not significant.

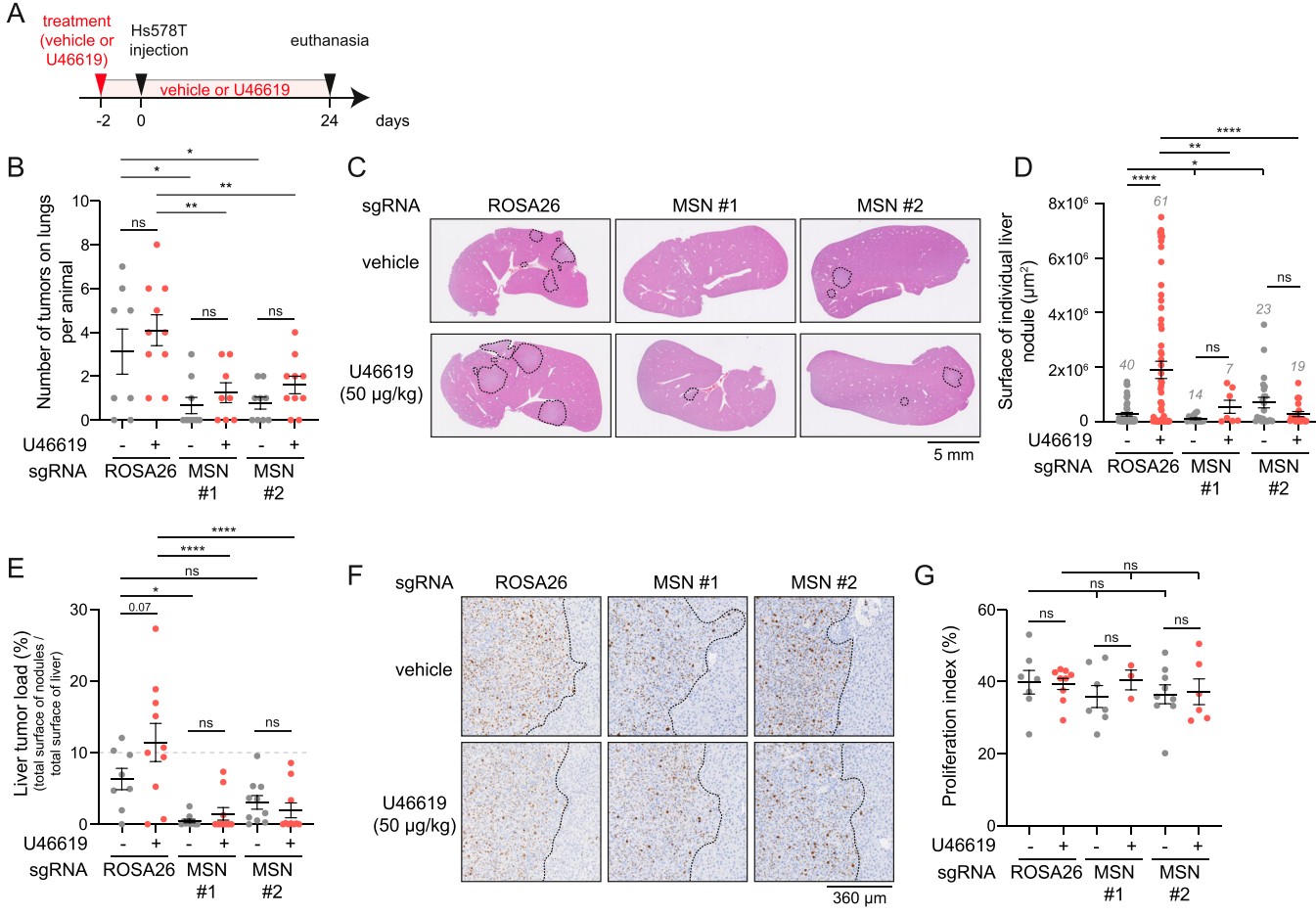

**Figure 6. TBXA2R-induced metastasis of triple-negative breast cancer cells is mediated by moesin in vivo.**
**(A)** Schematic representation of the in vivo tail vein–forced metastasis assay. NSG mice were injected with Hs578T cells after CRISPR-Cas9–mediated knockout for ROSA26 (control) or MSN using two independent sgRNAs. From 2 d before injection until euthanasia, mice were treated with vehicle or U46619 administered at 50 μg/kg through drinking water. n = 8 mice for ROSA26 + vehicle, and n = 10 mice for all other conditions. **(B)** Quantifications of the number of tumors on lungs per animal. Quantifications represent the mean ± s.e.m. of 8 or 10 animals. **(A, C, D, E)** Characterization of liver tumor nodules in mice treated as detailed in panel (A). **(C)** Representative H&E staining of liver sections. Black lines highlight tumor nodules. **(D, E)** Surface of individual tumor nodules was measured using NDP view 2 (Hamamatsu Photonics) on H&E-stained liver sections and represented as individual tumor nodule (D) or represented as a ratio of the total surface of nodules over total surface of liver per animal (expressed in %) (E). Quantifications represent the mean ± s.e.m. of 8 or 10 animals. **(A, F, G)** Representative IHC staining for Ki-67 in liver sections from mice described in (A). **(F)** Black lines demarcate the tumor (left) from the adjacent normal tissue (right) (F). **(G)** Proliferation index (Ki-67+ cells expressed in %) was then quantified (G). Quantifications represent the mean ± s.e.m of 8 or 10 animals. **(B, D, E, G)** Dots represent individual tumor nodules (D) or individual animals (B, E, G). **(B, D, E, G)** P-values were calculated using Welch's t test because of unequal sample sizes (D) or Holm–Sidak's multiple comparisons test with a single pooled variance (B, E, G). *P < 0.05, **P < 0.01, ****P < 0.0001, ns, not significant.

activates proteins of the ERM family, thereby enhancing cancer cell motility and invasion in vitro, as well as metastatic colonization in a mouse xenograft model. This signaling pathway is mediated through the Gα$_{q/11}$ and Gα$_{12/13}$ subfamilies, which both converge on GTPases of the Rho subfamily. In turn, these GTPases activate kinases of the SLK family that promote ERM activation.

Previous studies have identified ROCK, SLK, and LOK kinases as direct effectors of GTPases of the Rho subfamily (Matsui et al, 1996; Bagci et al, 2020) and activators of ERMs (Matsui et al, 1998; Belkina et al, 2009; Viswanatha et al, 2012; Machicoane et al, 2014). In our experiments with HEK293T, Hs578T, and MDA-MB-231 cells, we found that SLK and LOK, but not ROCK, are responsible for ERM activation, both under steady-state conditions and upon TBXA2R activation.

Recently, we reported that RhoA activates SLK to drive metaphase cell rounding by activating ERMs at the mitotic cortex (Leguay et al, 2022). We found that two different RhoGEFs, GEF-H1 and Ect2, activate RhoA at mitotic entry. Here, we showed that ERM activation depends on GTPases of the Rho subfamily downstream of Gα$_{q/11}$ and Gα$_{12/13}$ subfamilies. Although Gα$_{q/11}$ inhibition alone did not prevent ERM activation downstream of TBXA2R in HEK293T and TNBC cells, genetic depletion of Gα$_{12/13}$ partially prevented this activation. It is only when both Gα subfamilies are inhibited that TBXA2R signaling fails to activate ERMs. This indicates that whereas Gα$_{12/13}$ is the main Gα subfamily engaged by TBXA2R to activate ERMs, Gα$_{q/11}$ can partially compensate for the loss of Gα$_{12/13}$. Several RhoGEFs such as p63RhoGEF, p115RhoGEF, PDZ-RhoGEF, GEF-H1, and LARG (Sah et al, 2000; Lutz et al, 2007; Nakahata, 2008;

Wikstrom et al, 2008; Meiri et al, 2014; van Unen et al, 2016) were shown to be activated by Gα$_{12/13}$ or Gα$_{q/11}$ and represent possible links between these G proteins and GTPases of the Rho subfamily activation.

Although GPCRs and ERMs have been extensively studied, only a few studies have reported a link between these two protein families. For instance, stimulation of the muscarinic M1 receptor (M1R) or the β2-adrenergic receptor (β2AR) by their respective agonists was shown to promote ezrin activation through a mechanism involving the G protein–coupled receptor kinase 2 (Cant & Pitcher, 2005). Thrombin that activates PAR1, PAR3, and PAR4 was also reported to induce the recruitment of radixin at the plasma membrane of endothelial cells upon direct binding of this ERM to Gα$_{13}$ (Vaiskunaite et al, 2000). Lysophosphatidic acid (LPA), which activates LPARs$_{1-6}$, also promotes the activation of moesin in platelets (Nakamura et al, 1995; Shcherbina et al, 1999; Retzer & Essler, 2000). More recently, LPA activation of LPAR$_1$ and LPAR$_2$ was found to stimulate the phosphorylation of ERMs in the ovarian cancer cell line OVCAR-3 (Park et al, 2018). Interestingly, the expression of ezrin$^{T567A}$, a nonphosphorylatable mutant of the regulatory threonine that acts as a dominant-negative form of ERMs, reduced the migration of OVCAR-3 cells toward LPA, used as a chemoattractant.

Our study establishes the signaling cascade that connects a GPCR, TBXA2R, to ERM activation, which ultimately promotes TNBC cell migration, invasion, and metastatic colonization. Although it is currently unclear whether alternative mechanisms proposed for other GPCRs, such as the involvement of GRK2 (Cant & Pitcher, 2005) or direct binding of ERM to Gα$_{13}$ (Vaiskunaite et al, 2000), contribute to ERM activation downstream of TBXA2R, our findings indicate that SLK and LOK play a predominant role in mediating ERM activation in this context. This conclusion is based on functional studies conducted in three different cell lines and does not exclude the possibility that other kinases may contribute in distinct cellular or physiological settings. It is also unclear whether Gα$_{q/11}$ and/or Gα$_{12/13}$ are the only G proteins that mediate the activation of ERM downstream of the other GPCRs found to activate ERMs. However, it is noteworthy that M1R, β2AR, thrombin-activated PARs, and LPAR$_{1/2}$ were found to activate either Gα$_{q/11}$ or Gα$_{12/13}$ or both (Avet et al, 2022).

The molecular basis underlying metastasis is still incompletely understood, and the mechanisms that prompt cancer cells to disseminate to secondary organs are very diverse. These mechanisms depend on genetic alterations of cancer cells, but they also rely on the interaction of cancer cells with the stroma (Welch & Hurst, 2019). Thromboxane A2 (TXA2), the natural ligand of TBXA2R, was found to promote metastasis in colorectal (Guillem-Llobat et al, 2016), lung (Lucotti et al, 2019), prostate (Elliott et al, 2004), and breast (Ekambaram et al, 2011) cancers, opening the possibility that TBXA2R can be targeted to prevent metastasis. Supporting this hypothesis, ifetroban (CPI211), a potent and selective antagonist of TBXA2R, was recently shown to reduce the metastatic burden of several cancer cell lines, including TNBC cells, in xenograft mouse models (Werfel et al, 2020). Ifetroban is currently being evaluated in a clinical trial (NCT03694249) as a potential treatment for high-risk solid tumors that are prone to metastatic recurrence. The rationale of this trial is that ifetroban inhibits TBXA2R on platelets, which reduces their aggregation on cancer cells, ultimately preventing platelet-assisted metastasis (Lucotti et al, 2019; Werfel et al, 2020).

In addition to its role in cancer cell-intrinsic signaling, TXA$_2$-TBXA2R signaling has also been implicated in several non–cell-autonomous pro-metastatic processes. TXA2 has been shown to increase vascular permeability, promote platelet aggregation, and enhance inflammatory cell recruitment, different factors that collectively could facilitate tumor cell extravasation and metastatic seeding (Nie et al, 2000; Ashton et al, 2022; Liao et al, 2023; Xue et al, 2024). Our findings complement these non–cell-autonomous effects by indicating that TXA$_2$-TBXA2R signaling may contribute to metastasis both by directly enhancing tumor cell motility via ERM activation and by shaping a permissive metastatic niche.

Our study identifies a distinct mechanism by which TBXA2R could be targeted to block metastasis; activation of TBXA2R on TNBC cells can increase their motility, invasion, and metastatic potential by activating ERMs. The overexpression of essential components of this pathway could up-regulate this TBXA2R-ERM signaling axis. Supporting this notion, we observed in our TMA cohort that high TBXA2R expression was not detected in tumors with low levels of phosphorylated ERMs, a pattern consistent with a link between TBXA2R signaling and ERM activation. Because TBXA2R signaling is a potent activator of ERMs, activation of this GPCR by its natural ligand, TXA2, could also promote metastasis by activating ERMs.

Although our in vivo data support a functional role of TBXA2R-ERM signaling in metastatic progression, the use of a tail vein injection model focuses primarily on the later stages of metastasis and does not capture the full complexity of the metastatic cascade, including local invasion, intravasation, or interactions with the primary tumor microenvironment. Complementary approaches using orthotopic or spontaneous metastasis models will be valuable to further define the contribution of TBXA2R signaling to TNBC progression. The present study outlines a mechanistic rationale that can guide such extensions.

Identifying therapeutic targets for the treatment of metastatic disease is challenging because of its complexity, and so far, no effective antimetastatic drugs have been approved for clinical use (Anderson et al, 2019). However, the discovery that TBXA2R activates ERMs through Gα$_{12/13}$ and Gα$_{q/11}$ suggests that other GPCRs that engage one of these two Gα subfamilies could also promote metastasis by activating ERMs. Given that several GPCR antagonists are already approved drugs or have advanced to clinical trials for other indications, repurposing them for antimetastatic treatment could represent an appealing avenue that could accelerate the discovery of new antimetastatic treatments. Moreover, our characterization of the TBXA2R-ERM signaling pathway reveals novel potential therapeutic targets to prevent metastasis. Although ERMs are already promising therapeutic targets (Clucas & Valderrama, 2014; Ren & Khanna, 2014; Ghaffari et al, 2019; Hoskin et al, 2019), their activating kinases, SLK and LOK, which have not

been explicitly considered as targets yet, could also represent promising targets to prevent metastatic dissemination and outgrowth.

# Materials and Methods

### Reagents and inhibitors

Coelenterazine 400a (DeepBlueC) was purchased from NanoLight Technology (#340). U46619 and SQ-29548 were purchased from Cayman Chemical (#16450 and #19025, respectively). Calyculin A and Y-27632 ROCK inhibitors were purchased from Sigma-Aldrich (#C5552 and #688000, respectively). Staurosporine was purchased from APExBIO (#A8192). Rho inhibitor I (C3 transferase) was purchased from Cytoskeleton (#CT04). YM-254890 was purchased from FUJIFILM Wako Pure Chemical Corporation (#257-00631).

### DNA constructs

The cDNA for TBXA2R was previously described (Parent et al, 1999). (MyrPB-)ezrin-, radixin-, moesin-rLucII, and rGFP-CAAX constructs were previously described (Leguay et al, 2021). MISSION shRNA constructs were obtained in pLKO.1-puro vectors from Sigma-Aldrich: EZR (TRCN0000062459), RDX (TRCN0000062434), MSN (TRCN0000062412). FlexiTube siRNAs were obtained from QIAGEN: SLK#1 (SI00107723), SLK#2 (SI04438350), LOK (SI02224054).

### Cell culture and transfection

HEK293T human kidney cells, and Hs578T and MDA-MB-231 TNBC cells were cultured in DMEM (4.5 g/liter D-glucose, L-glutamine, 110 mg/liter sodium pyruvate; #11995073; Thermo Fisher Scientific), and MB-468 and BT-549 TNBC cells in RPMI 1640 medium (Invitrogen) at 37°C with 5% $CO_2$. DMEM and RPMI were supplemented with 10% FBS (#12483020; Life Invitrogen) and 1% penicillin–streptomycin antibiotics (#15070063; Thermo Fisher Scientific). MDA-MB-157 cells were cultured in Leibovitz's L-15 medium (Cedarlane), supplemented with 15% FBS and 1% penicillin–streptomycin, and maintained at 37°C without $CO_2$.

HEK293T cells knocked out for $G_{q/11}$ or $G_{12/13}$ or every Gα subunit were obtained from Asuka Inoue (Tohoku University, Japan). Hs578T cell clones knocked out for ezrin, radixin, moesin, or SLK/LOK were obtained by CRISPR-Cas9, followed by clonal dilution after selection with 2 μg/ml puromycin (#540222; EMD Millipore) for 48 h. The following sequence guides were inserted in plentiCRISPR-v2 (#52961; Addgene) digested with BsmBI (#R0739; New England Biolabs): EZR (CTGAGCGGCTGATCCCTCAA), RDX (GTTTCACTTACCGCTGGGGT), MSN#1 (GAGACAAGTTGCTCCCGCAG), MSN#2 (AAGCTTACCTGAGCATGCCA), SLK (CAATTTGATATCTCCATCTA), and LOK (CATGATTGAGTTCTGTCCAG).

For ebBRET experiments, HEK293T cells were transfected using linear polyethyleneimine (PEI, #43896; Alfa Aesar) as previously described (Leguay et al, 2021). When different conditions were compared, cells were reverse-transfected in suspension before splitting and plating, ensuring that all conditions expressed similar TBXA2R levels within the same experiment. In rescue experiments,

Gαq and Gα13 were cotransfected with biosensors and receptors. Loss-of-function experiments were performed by transfecting siRNA using Lipofectamine RNAiMAX (#13778075; Thermo Fisher Scientific) as prescribed by the manufacturer or by infecting cells with lentiviral particles in DMEM supplemented with 10% FBS and 5 μg/ml polybrene (#H9268; Sigma-Aldrich) for 48 h followed by a 48-h treatment with 2 μg/ml puromycin.

For sequential treatments of different durations, the longest treatment was initiated first to ensure that all treatments concluded simultaneously; the indicated times correspond to the duration of each treatment before cell lysis.

### ebBRET measurement

ebBRET measurements were performed as previously described (Leguay et al, 2021). Briefly, 48 h after transfection, HEK293T cells were washed with Hanks' balanced salt solution (HBSS, #14065056; Thermo Fisher Scientific) and incubated for 5 min with 2.5 μM coelenterazine 400a diluted in HBSS. ebBRET signals were monitored using a Tecan Infinite 200 PRO multifunctional microplate reader (Tecan) equipped with BLUE1 (370–480 nm; donor) and GREEN1 (520–570 nm; acceptor) filters. ebBRET signals were calculated as a ratio by dividing the acceptor emission value by the donor emission value.

### Immunoblotting

After the indicated treatment, cells were washed with ice-cold PBS and lysed in TLB buffer (40 mM Hepes, 1 mM EDTA, 120 mM NaCl, 10 mM NaPPi, 10% glycerol, 1% Triton X-100, 0.1% SDS) supplemented with both phosphatase and protease inhibitors (phosphatase inhibitor cocktail [PIC, #P2850; Sigma-Aldrich], 1 mM sodium orthovanadate [$Na_3VO_4$, #S6508; Sigma-Aldrich], 5 mM β-glycerophosphate [#G6251; Sigma-Aldrich], 1 mM phenylmethylsulfonyl fluoride [PMSF, #P7626; Sigma-Aldrich], and anti-protease cocktail [#4693132001; Sigma-Aldrich]). Samples were then denatured in sample buffer (200 mM Tris–HCl 1 M, pH 6.8, 8% SDS, 0.4% bromophenol blue, 40% glycerol, and 412 mM β-mercaptoethanol) and resolved by SDS–PAGE followed by transfer to nitrocellulose membranes (pore 0.2 μm, VWR #27376-991). Membranes were blocked in TBS–Tween (25 mM Tris–HCl, pH 8, 125 mM NaCl, 0.1% Tween-20) supplemented with 2% BSA for 1 h before overnight incubation with primary antibodies at 4°C. Primary antibodies were as follows: rabbit anti-p-ERM (1:5,000; Roubinet et al, 2011), rabbit anti-ERM (1:1,000, #3142; Cell Signaling), rabbit anti-ezrin (1:1,000, #3145; Cell Signaling), rabbit anti-radixin (1: 1,000, #2636; Cell Signaling), rabbit anti-moesin (1:1,000, #3150; Cell Signaling), mouse anti-actin (1:5,000, #MAB1501; Sigma-Aldrich), rabbit anti-SLK (1:1,000, #A300-499A; Cedarlane), and rabbit anti-LOK (1:5,000, #ab70484; Abcam). Washed membranes were then incubated for 1 h at room temperature with secondary antibodies: goat anti-rabbit-IgG HRP (1:10,000, #sc-2004; Santa Cruz Biotechnology) and goat anti-mouse-IgG HRP (1:10,000, #sc-516102; Santa Cruz Biotechnology). Protein detection was finally performed using Amersham ECL Western blotting detection reagent (#CA95038-564L; GE Healthcare). Note that the total ERM antibodies recognize non-phosphorylated ERM but may recognize a fraction of phosphorylated ERM, potentially in a nonlinear manner. Nevertheless, normalization

of p-ERM to total ERM remains the most appropriate approach to quantify ERM activation, provided that total ERM levels are stable or experimentally controlled, and complementary actin normalization is shown when needed to document equal loading.

## In vitro kinase assay

In vitro kinase assays were performed as previously described (Leguay et al, 2022). Briefly, HEK293T cells treated with U46619 or its vehicle were lysed in TLB buffer supplemented with phosphatase and protease inhibitors. Endogenous SLK was then immunoprecipitated from the cell lysate using rabbit anti-SLK antibody (#A300-499A; Bethyl Laboratories) incubated for 1 h at 4°C, followed by incubation with protein A Sepharose beads (#GE17-0780-01; GE Healthcare) for 2 h at 4°C. Beads were then extensively washed and resuspended in kinase reaction buffer (KRB; 50 mM Tris–HCl, pH 7.5, 100 mM NaCl, 6 mM $MgCl_2$, and 1 mM $MnCl_2$) supplemented with both phosphatase and protease inhibitors, 2 mM DTT, 50 $\mu$M ATP, and purified recombinant GST-ezrin$^{479-585}$ from BL21 bacteria. The kinase assay was finally performed for 30 min at 30°C, and proteins were denatured with sample buffer.

## Immunofluorescence

The day before the experiment, cells were plated on glass coverslips (#0115200; Marienfeld) and incubated at 37°C with 5% $CO_2$ overnight. Cells were then washed with PBS and fixed with 10% trichloroacetic acid (#T0699; Sigma-Aldrich) for 10 min at room temperature, followed by extensive washes with TBS (20 mM Tris–HCl, pH 7.5, 154 mM NaCl, 2 mM EGTA, 2 mM $MgCl_2$). Cells were then permeabilized with 0.02% saponin (#0163; Amresco) diluted in TBS supplemented with 2% BSA (#ALB001.250; BioShop) for 1 h. Cells were then incubated overnight with rabbit anti-p-ERM primary antibody (1:5,000; Roubinet et al, 2011) or anti-p-myosin II antibody (1:200, #3671; Cell Signaling Technology) followed by goat anti-rabbit Alexa Fluor 488–conjugated secondary antibody (1:200, #A11070; Invitrogen) for 1 h. Glass coverslips were finally mounted in Vectashield medium with DAPI (#H-1200; Vector Laboratories), and images were acquired using an LSM 700 confocal microscope (Zeiss) with a 63x objective.

## Cell migration and invasion assay

Migration experiments were performed using the Oris Cell Migration assay (Platypus Technologies) as the manufacturer prescribed. Briefly, Hs578T cells were plated in DMEM supplemented with 10% FBS in a 96-well plate with stoppers creating a central cell-free detection zone. The next day, stoppers were removed, and media were replaced with DMEM supplemented with 10% FBS and 10 nM U46619 or vehicle. Cells were then allowed to migrate into the free area.

Invasion experiments were performed using the Oris Cell Invasion assay (Platypus Technologies) as prescribed by the manufacturer. Briefly, Hs578T cells were plated/embedded in DMEM supplemented with 2 mg/ml collagen type I (rat tail) (#CB354249; Thermo Fisher Scientific) in a 96-well plate with stoppers creating a central cell-free detection zone. After collagen polymerization, stoppers were removed, and DMEM supplemented with 10% FBS

and 2 mg/ml collagen type I was added to the central cell-free detection zone. After collagen polymerization, DMEM supplemented with 10 nM U46619 or vehicle was added on the top of collagen. Cells could then invade the free area.

Migration and invasion of cells into the free zone were recorded by contrast video microscopy using a LSM 700 confocal microscope (Zeiss) with 20x objective for 6 or 24 h, respectively. Individual cells were manually tracked using the Manual Tracking plugin on ImageJ software (NIH). Cell velocity was determined as a ratio by dividing the total distance v by time. Representations of cell migration were obtained using Chemotaxis and Migration Tool 2.0 (ibidi).

## Immunohistochemistry

Tissue microarray was obtained as previously described (Yousef et al, 2014). Samples were obtained from the *Centre Hospitalier de l'Université de Montréal* (CHUM) after approval by the research ethical committee (Comité d'éthique de la recherche du CHUM CENTRE DE RECHERCHE, Approval No. SL 05.019).

Livers excised from mice were fixed in 10% formalin, embedded in paraffin, and sliced into 4-$\mu$m sections. Tissue sections were mounted on glass slides and stained with hematoxylin and eosin (H&E) using conventional protocols. Immunohistochemical assays were performed on Bond RX Stainer (Leica Biosystems) according to the manufacturer's instructions. Briefly, slides were deparaffinized, and antigen recovery was conducted by heat-induced epitope retrieval with standard CC1 and CC2 solutions (Ventana Medical Systems). Sections were then incubated with the primary antibody for 30 min. Primary antibodies used are the following: p-ERM (1:600 [Roubinet et al, 2011]), TBXA2R (1:100, #10004452; Cayman chemicals), and Ki-67 (1:150, #CRM325B; Biocare Medical). Bound primary antibodies were detected using peroxidase polymer (HRP-DAB) for 15 min. The stained slides were next subjected to digital slide scanning that converts glass slides into high-resolution digital data by high-speed scanning using the NanoZoomer Digital Pathology (NDP) 2.0-HT digital slide scanner (Hamamatsu). Signal intensities for p-ERM and TBXA2R were quantified at the plasma membrane using ImageJ software (NIH). Quantification of Ki-67–positive cells was performed using VisioMorph software (Visiopharm).

## In vivo metastasis assay

Hs578T cells (5.0 × 10$^4$) were resuspended in 100 $\mu$l PBS and injected into the tail vein of NSG mice. Two days before cell injection and during the whole experiment, NSG mice were treated with U46619 (50 $\mu$g/kg) or vehicle (DMSO) administered through drinking water containing 0.1% aspartame (#A0997; TCI chemicals), given ad libitum and changed every second day. The mice were then euthanized 24 d after cell injection in a $CO_2$ chamber. At the time of euthanasia, the liver was photographed and weighed, and the number of tumors was manually assessed under a stereo microscope. The left lobe was then processed for histological analysis.

## Cell proliferation assay

Cell proliferation was measured using MTT. Briefly, Hs578T cell lines were seeded into 96-well plates in quadruplicate and cultured for

the durations indicated in the figure legend. The medium was then replaced with fresh medium supplemented with 1.1 mM MTT (#21795; Cayman Chemical), and cells were incubated at 37°C for 4 h. Dissolution of formazan was then performed by incubating cells with 67% DMSO at 37°C for 10 min. Cell proliferation was assessed by reading the absorbance at 540 nm using a Tecan Infinite 200 PRO multifunctional microplate reader (Tecan).

### Data analysis

All quantifications were performed using ImageJ software (NIH) and analyzed using GraphPad PRISM software (GraphPad Software). Microscopy images were prepared using ImageJ software (NIH) and Photoshop (Adobe).

For cancer survival analysis, we used gene expression data from TNBC and normal breast tissues obtained from The Cancer Genome Atlas (TCGA). TCGA-BRCA RNA-seq dataset (mRNA_Preprocess_Median) was downloaded from the Broad GDAC Firehose (https://gdac.broadinstitute.org/). For survival analyses, we calculated the mean expression of *TBXA2R* and *MSN* and categorized the samples into groups based on the upper and lower tertiles. Overall survival was analyzed using the Kaplan–Meier method, and differences in survival rates were compared using the log-rank test in GraphPad PRISM software.

# Supplementary Information

# Acknowledgements

This work has been supported by a Project Grant from the CIHR (175193) to S Carréno and M Bouvier, a Foundation Grant from the CIHR (148431) to M Bouvier, and a Cancer Research Society Grant (25388) to S Meloche. K Leguay held a doctoral scholarship from the Institute for Research in Immunology and Cancer (IRIC), from Montreal University's Molecular Biology Program, from Études Supérieures et Postdoctorales, Montreal University, and from the Fonds de recherche Santé Québec (FRQS). O Naffati holds a master's and doctoral scholarship from the Mission Universitaire de Tunisie en Amérique du Nord, from Montreal University's Molecular Biology Program, and from IRIC. CL Kiyan holds a doctoral scholarship from the Institute for Research in Immunology and Cancer (IRIC) and from Montreal University's Molecular Biology Program, as well as a Merit scholarship from the Faculty of Medicine of the University of Montreal. C Tesnière held a doctoral studentship from the FRQS. M Bouvier held the Canada Research Chair in Signal Transduction and Molecular Pharmacology. The authors are grateful to Dr. Asuka Inoue for providing the $\Delta G\alpha_{q/11}$, $\Delta G\alpha_{12/13}$, and $\Delta G\alpha$ HEK293T cells.

### Author Contributions

K Leguay: conceptualization, data curation, formal analysis, supervision, validation, investigation, visualization, methodology, and writing—original draft, review, and editing.
O Naffati: conceptualization, data curation, formal analysis, validation, investigation, visualization, and methodology.

CL Kiyan: data curation, formal analysis, validation, investigation, visualization, and methodology.
YY He: formal analysis, investigation, and methodology.
M Hogue: investigation and methodology.
C Tesnière: formal analysis, investigation, and methodology.
M Gombos: formal analysis and investigation.
H Kuasne: data curation, formal analysis, investigation, visualization, and methodology.
L Gaboury: formal analysis, investigation, and methodology.
C Le Gouill: conceptualization, formal analysis, and methodology.
S Meloche: conceptualization, resources, funding acquisition, methodology, project administration, and writing—original draft, review, and editing.
M Bouvier: conceptualization, resources, data curation, formal analysis, supervision, funding acquisition, validation, methodology, project administration, and writing—original draft, review, and editing.
S Carréno: conceptualization, resources, data curation, formal analysis, supervision, funding acquisition, visualization, methodology, project administration, and writing—original draft, review, and editing.

### Conflict of Interest Statement

The authors declare that they have no conflict of interest.

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
