## [Reviewer comments · Life Science Alliance]

TBXA2R activates ERMs to drive motility, invasion and metastatic colonization of TNBC cells.

Kévin Leguay, Omaira Naffati, Camila Lie Kiyon, Yu Yan He, Mireille Rogue, Chloe Tesniere, Melania Gombos, Hellen Kuasne, Louis Gaboury, Christian Le Gouill, Sylvain Meloche, Michel Bouvier and Sébastien Carréno

DOI: <https://doi.org/10.26508/lsa.202503601>

Corresponding author(s): Prof. Sébastien Carréno (Université de Montréal)

Review Timeline:	Submission Date:	2025-12-15
	Editorial Decision:	2026-02-06
	Revision Received:	2026-02-24
	Accepted:	2026-03-03

Scientific Editor: Tim Fessenden

Transaction Report:

Please note that the manuscript was reviewed at *Review Commons* and these reports were taken into account in the decision-making process at *Life Science Alliance*.

Reviews

Review #1

1. Evidence, reproducibility and clarity:

****Summary****

This manuscript investigates the role of the thromboxane A2 receptor (TBXA2R) in activating ERM (ezrin, radixin, and moesin) proteins to promote cell motility and invasion in triple-negative breast cancer (TNBC) cells. Using TBXA2R stimulation and a series of in vitro and in vivo experiments, the authors report that ERM activation is mediated through a TBXA2R signaling pathway involving Gαq/11 and Gα12/13 subunits, RhoA, and SLK/LOK kinases. They propose that this pathway enhances cell migration, invasion, and metastatic potential in TNBC.

****General criticisms****

Experimental design and analyses are adequate, even though certain experiments lack appropriate controls or employ the wrong statistical tests. However, the study primarily relies on a single TNBC cell line and heavy use of overexpression systems and/or small molecule inhibitors, raising concerns about the generalizability and specificity of the findings. Furthermore, several conclusions appear premature and unsupported by the current data. Critical controls and additional validation experiments are necessary to support the claims about the role of TBXA2R in metastasis and to justify the strong mechanistic conclusions drawn.

****Specific criticisms****

Figure 1

TBXA2R expression should be shown to understand whether different ebBRET signals are dependent on the overexpression levels of TBXA2R.

E-F: As ERM levels change over time, one would like to understand whether this is due to misloading or whether there is an underlying biological event going on in the stimulated cells. Are total ERM levels really changing over time? Please add a blot for 1-2 housekeeping proteins as loading controls. This is also crucial to clarify the kinetics of ERM activation; such notable intensity variations make quantifications of non-linear WB signals not fully reliable. In F, mean and SD should be plotted.

G: The authors need to use a PM marker if they want to claim that pERM increases at the cell cortex. TBXA2R localization should also be shown.

Figure 2

A: This reviewer cannot see the purported partial inhibition in Gα12/13 KO cells. Are differences between the two KOs significant? Furthermore, there are reports indicating that YM-254890 may not be specific for Gαq. Experiments on double KO cells are needed to assess the possible redundancy between the two Gα subfamilies.

C-D: it is important to add a positive control for the activity of Y-27632 in these experiments. Please show that a ROCK-dependent effect is inhibited in the treated cells.

G: The working model is premature as it is unknown whether ROCKi was active. While asking for ROCK1/2 KO cells would be too much, this claim is far-fetched.

Figure 3

B: In the legend, it is not clear what grey and light read colours mark.

E-F: This reviewer finds it difficult to believe that p-ERM and TBXA2R signal intensities at the cell cortex could be reliably quantified using IHC images. The representative samples would indicate that p-ERM and TBXA2R positivity are not correlated. It would be crucial to show examples for each of the TNBC subgroups the existence of which is inferred based on p-ERM and TBXA2R staining.

The conclusion that "no TNBC samples exhibited high TBXA2R expression and low levels of p-ERMs, further supporting a role for TBXA2R signalling in ERM activation in TNBC" is an overstatement.

Figure 4

The authors wrote that "We focused on the Hs578T cell line, which showed a median level of TBXA2R mRNA expression among the six TNBC cell lines tested". I do not understand the rationale for it as anti-TBXA2R antibodies detecting endogenous TBXA2R are available and thus why not use the median protein levels?

Figure 5

Effects of the knockouts are subtle, and rescue experiments would be needed to corroborate these results. The employed statistical analysis is prone to overestimating differences. The authors should use the superplots instead. The authors might also decide to use other TNBC cell lines to explore the functional relevance of this pathway in BC progression. This is particularly important because Hs578T are poorly tumorigenic, and they often do not form palpable tumours in mice.

Figure 6

The fact that Hs578T are poorly tumorigenic in mice is likely the reason why the authors used the experimental metastasis model. However, it is puzzling that metastases were studied in the liver but not in the lungs. Furthermore, the whole approach is rather artefactual as the TBXA2R agonist was administered for the entire duration of these experiments. What is the pathological relevance of such a study?

Including a spontaneous metastasis model or alternative TNBC lines that mimic human disease more closely would help strengthen the functional relevance of this pathway in BC progression and study's translational relevance.

Figure S2

B-M: the pERM signal appears to be perinuclear in some of the tested cell lines. Please use a PM marker.

Figure S3

The authors should use the superplots to analyse the cell migration data.

Discussion

The claim that "our findings demonstrated that kinases of the SLK family are the only kinases needed for ERM activation by TBXA2R" should be tuned down as only 2 cell lines were tested. In this section, the authors should also discuss the proposed pro-metastatic functions of TXA2 and TXA2R in more detail, including vascular permeability. The sweeping conclusion that "TBXA2R expression correlates with phosphorylation and activation of ERMs in TNBC patient samples" clashes with the authors' own results; please stick to the data.

Concluding remarks

This study investigates a signaling pathway whereby TBXA2R through ERM activation enhances the migratory and invasive potential of TNBC cells. However, several improvements are needed to support the main claims. The dependence on a single TNBC cell line, reliance on pharmacological inhibitors with potential off-target effects, and limited in vivo relevance detract from the generalizability of the findings. Additional TNBC models, adequate controls, and a broader focus on natural metastasis patterns would make the conclusions more compelling. Moderating certain overstated claims would be needed to align the interpretations with the actual data.

****Cross-commenting****

I found comments in the other reviewers' reports that align with my criticisms on the mouse experiments as well as with those pertaining to the tissue culture work.

2. Significance:

Significance (Required)

General comments

The manuscript investigates the role of TBXA2R in the regulation of ERM in the context of TNBC metastasis. Much of this TBXA2R signalling axis is already known, as well as that SLK and LOK can phosphorylate ERM in other cell systems. Similarly, the positive role of ERM in cell migration/invasion and cancer progression has long been reported. The somewhat unexpected finding that ERM phosphorylation is independent of ROCK remains not fully convincing. The BC-related part is problematic as the continuous administration a TBXA2R agonist is required for key tumour metrics to show some differences in vivo. This calls into question the main conclusion of the work, namely that the TBXA2R/ERM-dependent pathway is activated during BC progression in TNBC cells.

Audience

Specialists interested in GPCRs and signal transduction or in the cytoskeleton.

Expertise

Rev: cancer cell biology, signal transduction, cytoskeleton, actin biochemistry, multiplexed imaging, mouse model of human diseases.

Co-rev: nanoparticles, cell biology.

3. How much time do you estimate the authors will need to complete the suggested revisions:

Estimated time to Complete Revisions (Required)

(Decision Recommendation)

More than 6 months

4. Review Commons values the work of reviewers and encourages them to get credit for their work. Select 'Yes' below to register your reviewing activity at Web of Science Reviewer Recognition Service (formerly Publons); note that the content of your review will not be visible on Web of Science.

Yes

Review #2

1. Evidence, reproducibility and clarity:

Leguay et al present an interesting and logical series studies that investigate the activity and signaling of the GPCR TBXA2R in TNBC cells. The premise of the overall study is that metastasis is often associated with a more invasive/motile cancer cell phenotype. The investigators have an interest in ERM (Ezrin, Radixin, Moesin) proteins, which have been implicated in cell motility. The authors link stimulation of TBXAR2, a GPCR, to activation of ERM proteins and also show that TBXAR2 is associated with worse outcome in TNBC patients. Through the use of genetic and pharmacologic tools the authors provide convincing biochemical and cell based data to support their model that stimulation of TBXAR2 activates Gα11 & Gα12/13 which subsequently stimulate RhoA and SLK/LOK which then phosphorylate ERMs. The authors show relevant biologic consequences of the pathway. Data include orthogonal assays with similar results and the manuscript is written clearly and the data are displayed well. Overall it is a solid story that is largely well done. There are a few comments that should be addressed.

****Comments:****

1. All the biochemical/cell based in vitro data exploit the use of small molecule agonists of TBXAR2, not the natural ligand. A comment on this and why use of TXA2 is not feasible would be helpful to the reader.
2. The data in figures 1-5 are solid and clear. However, I suggest adding a higher magnification inset for the IHC images shown in Fig 3E. It would be useful to be able to distinguish cells in the IHC, a higher mag shot should suffice.
3. A) The use of Hs578t cells for the in vivo modeling is unfortunate. Additionally, the use of iv injection to in a study focused on cell invasion is also unfortunate. The metastatic propensity of Hs578t is not clear, in fact a recent report comparing metastasis in breast cancer cell lines shows that Hs578t perform poorly in terms of metastasis after orthotopic injection (see PMID 38468326). I searched the literature a bit to try and find other examples of iv injection of Hs578t cells, I found 1 (PMID:27654855, I did not search exhaustively), this paper shows significant lung metastasis

and does not mention liver metastases. Were other breast cancer cells investigated for the in vivo studies?

B) Why I was interested is because the typical organ that is seeded post iv injection is the lungs (as seen in the above ref), liver metastases post iv injection are not common, especially with breast cancer cells. What did the lungs look like in your experiments?

C) Further while the data presented in figure 6 are supportive of the overall conclusions, the data is modest at best in terms of metastatic burden. Repetition of the experiment using a breast cancer cell line injected orthotopically would likely be more useful in highlighting the importance of the pathway to metastasis.

I understand performing an orthotopic assay may be outside the scope of the study, but it would provide greater impact given the focus of the paper on cell invasion.

****Cross-commenting****

I think reviewer comments are generally aligned. I was least critical but appreciate the concerns of the other reviewers, especially rev #1 who requested additional validation and controls. In my opinion in vivo studies are not robust, I expect that is due to cell line choice. Repetition of the in vivo study with a breast cancer cell line that is capable of metastasis (from a primary tumor) would be more effective.

2. Significance:

Significance (Required)

The manuscript presents a solid, logical flow and the biochemical/cell based in vitro data are clean. Clear differences between groups, appropriate controls, and displayed effectively.

The challenge is the in vivo study. IV injection of cancer cells is a valid model for seeding and growing in a target organ BUT it does not reflect cell invasion, which is typically thought of as a step that occurs earlier in the metastatic cascade.

That said, the data are supportive with conclusions but not necessarily consistent with expected results based on iv injection of this cell line. A caveat is that the cell line used is characterized as having metastatic characteristics in vitro but is not a consistent metastatic line in vivo.

The recommendation is to perform a new in vivo experiment. An orthotopic injection of a strongly metastatic cell line, such as MDA MB 231 or other (see paper ref above) would be a more stringent and accurate test of the importance of the pathway to cell invasion in vivo.

3. How much time do you estimate the authors will need to complete the suggested revisions:

Estimated time to Complete Revisions (Required)

(Decision Recommendation)

Between 3 and 6 months

4. Review Commons values the work of reviewers and encourages them to get credit for their work. Select 'Yes' below to register your reviewing activity at Web of Science Reviewer Recognition Service (formerly Publons); note that the content of your review will not be visible on Web of Science.

No

Review #3

1. Evidence, reproducibility and clarity:

****Summary****

The Ezrin, radixin, and moesin (ERM) family of proteins orchestrate morphological changes that potentiate metastatic invasion in cancer cells. In this study, Leguay et al. identify the GPCR, TBXA2R, as a key activator of the ERM proteins which promotes motility and invasion in triple-negative breast cancer (TNBC) cells. Using BRET-based sensors developed by them previously for monitoring the activation of ERM proteins and building upon their previous findings on the role of the small GTPase RhoA in the activation of ERM proteins, the authors carefully dissect the

molecular pathway leading to the activation of ERM proteins upon stimulation of the TBX2AR. The authors also establish the pathological relevance of the pathway in TNBC using in vitro and in vivo models, opening up possibilities for targeting this pathway in cancer cells. Overall, the study is well-conceived and executed, and the results are clearly described and presented in the manuscript. However, the following comments must be addressed before publication.

****Major comments****

Fig 1C - Why p-ERM was normalized over Ezrin and not ERM? It would be more appropriate and consistent to normalize against the ERM signal as done in other experiments in the manuscript.

Fig 1E and S3C - The levels of total ERM also seem to change with increasing treatment times. This must be clarified and discussed in the manuscript.

Fig 1F - Why is the mean of all three independent experiments not presented here as in S3C?

Fig 2E - Though SLK seems to play a dominant role in the phosphorylation of ERM in HEK293T cells, the depletion of LOK also substantially reduces the phosphorylation of ERM in the representative figure (Fig 2E), which is not reflected in the quantification (Fig 2F). Indeed, both SLK and LOK seem to be equally crucial in Hs578T cells (Fig 4I), unlike the conclusion here. The authors must check if the quantifications were affected by any white spots in the blot for total ERM as seen in the representative figure. If necessary, the authors must include additional replicates, and the model in Fig 2G should be updated accordingly. If the contributions of LOK are indeed quite minimal in HEK293T cells, then the difference in Hs578T cells must be adequately highlighted and discussed rather than broadly mentioning similar results were observed in both cell lines.

The discussion mentions that SLK kinases are the only kinases needed for ERM activation, which conflicts with findings from Hs578T cells, where both SLK and LOK contribute to ERM phosphorylation (Fig 4I). The authors should revise this to reflect their data accurately.

****Minor comments****

FigS3B should cite the source dataset and not just the database. Also, details of how the extracted data was processed (if any) should be described clearly.

When multiple treatments are involved (for, e.g. U46619 and staurosporine), the exact sequence of treatments and the overlap in timings of different treatments must be clearly mentioned. E.g. fig 1A and 1C.

There are a few grammatical errors which need to be fixed. E.g. Paragraph 2 in the second section of results - We next aimed to identify (not identifying) which kinase(s) acts downstream of TBX2AR

2. Significance:

Significance (Required)

Triple-negative breast cancer, which is characterized by a lack of estrogen, progesterone or HER2 receptors, is a highly metastatic and aggressive form of breast cancer with poor prognosis. Currently, there are fewer treatment options than other types of invasive breast cancer. The current study opens up the possibility of targeting the TBX2AR or the downstream signalling components in TNBC, which are still expressed in TNBC cells. However, certain TNBC sub-types express low levels of p-ERM and TBX2AR (Fig 3E, 3F), indicating a minor role for TBX2AR pathway and targeting this pathway in these subtypes may be inefficient. In addition, certain subtypes express high p-ERM and low TBX2AR indicating alternative pathways for ERM activation. Currently, it is not clear which other GPCRs can contribute to ERM activation by engaging similar downstream effectors. A comprehensive screening of different GPCR antagonists could identify alternative strategies to target the ERM-mediated metastasis in TNBC cells that show low expression of TBX2AR.

Audience The manuscript is relevant to a broad audience, especially to cell biologists, cancer biologists and clinical scientists.

The reviewer's field of expertise includes cell signaling, gene expression, and RNA biology in mammalian systems. Moderate expertise in cancer biology. Limited knowledge of histopathological analysis.

3. How much time do you estimate the authors will need to complete the suggested revisions:

Estimated time to Complete Revisions (Required)

(Decision Recommendation)

Less than 1 month

4. Review Commons values the work of reviewers and encourages them to get credit for their work. Select 'Yes' below to register your reviewing activity at Web of Science Reviewer Recognition Service (formerly Publons); note that the content of your review will not be visible on Web of Science.

Yes

Review #4

1. Evidence, reproducibility and clarity:

Overall, the authors show an interesting and conclusive work on the activation of ERM proteins upon TBXA2R signaling. The use of the ebBRET biosensor to assess ERM-protein activation enables elegant investigation of activation modalities. The Thromboxane A2 analogue U46619 robustly shows activation of ERM proteins in ebBRET assays as well as an increase in ERM-protein phosphorylation status. The functional effects of this signaling pathway are shown convincingly for moesin, where moesin mediates an TBXA2R mediated increase in cell motility, invasion and metastasis of triple-negative breast cancer Hs578 cells in vitro and in vivo. Nonetheless, some points need to be clarified.

2. Significance:

Significance (Required)

Comment 1: In the title the authors state, that ERM-activation via TBXA2R is controlling invasion and motility of triple-negative breast cancer cells. In the manuscript, there is only data supporting this assumption for moesin (MSN). Therefore, the authors need to change the title accordingly or support additional experiments for the other two ERM-proteins radixin and ezrin. Throughout the experiments, the p-ERM antibody is used to measure ERM-protein activation. Since the effects on invasion and motility observed in Hs578 cells are mainly mediated through moesin, it would be necessary to see, at least for one experiment per cell line (HEK293T, Hs578) the detailed phosphorylation status of ezrin, radixin and moesin separately. As there are specific, phospho-detecting antibodies for this case, this could be done rather easy. Furthermore, showing specific increase of phosphorylated moesin would support the functional data shown in Figure 5 and 6. To investigate the functional effect of TBXA2R mediated activation of ezrin and radixin on cell motility and invasion, similar experiments could be done in e.g. HMC-1-8 breast cancer cells (high ezrin expression) and HCC1187 (high radixin expression).

Comment 2: Figure 1A, C, D: The concentration of staurosporine is with 100 nM relatively high for kinase inhibition. It would be informative to see the assay with increasing staurosporine concentrations, e.g. from 1 nM to 50 nM. In general, a concentration of 1-10 nM should be sufficient for kinase inhibition, preventing unspecific effects of the drug.

Comment 3: The citation for the p-ERM antibody is confusing, as there is only p-Moe used in the cited paper (Roubinet, 2011). There is a p-ERM antibody commercially available (Cell Signaling, Phospho-ezrin (Thr567)/radixin (Thr564)/moesin (Thr558) Antibody #3141). Could you clarify which antibody you are using?

Comment 4: From the inhibitor experiments using C3 transferase toxin (Figure 2), the authors conclude that RhoA plays a role in TBXA2R mediated ERM activation. As mentioned in the manufacturer's description, C3 toxin is inhibiting RhoA, RhoB and RhoC. Therefore, it would be necessary to repeat those experiments under RhoA knockdown conditions (e.g. using an siRNA-based approach) to state that specifically RhoA is involved.

Comment 5: To assess, if the findings in Figure 5 and 6 are due to the higher moesin expression in Hs578 cells or are linked to a specific function of moesin, a re-expression experiment would be informative. To achieve this, the 2D and 3D migration experiments could be redone after re-expression of moesin, ezrin and radixin separately in moesin knockdown conditions.

Minor comments:

- Even though U46619 is a known Thromboxane A2 analogue, including negative and positive controls would strengthen the results. In detail, this could be done by showing a known protein which gets phosphorylated downstream of TBXA2R signaling and a protein which is not affected by this signaling pathway alongside the shown effects on ERM-proteins.
- Figure 1 J: There are no statistics comparing the conditions of SQ-29548 treated cells in presence/absence of U46619, that should be added.
- Figure 1 G, H: How was the quantification for cell periphery performed? In detail, how were the thresholds set for cell periphery / not cell periphery?
- Figure 3 H:
 - The labelling indicating presence of U46619 is missing.
 - Also, what is the rationale behind normalizing MB-453 for 3 cell lines and comparing the BT-549 to MB-157?
- Suppl. Fig 4 D: Define y-axis better. Absorbance at what wave length?
- Define FERM and ERMAD abbreviations in introduction.

3. How much time do you estimate the authors will need to complete the suggested revisions:

Estimated time to Complete Revisions (Required)

(Decision Recommendation)

Between 3 and 6 months

4. Review Commons values the work of reviewers and encourages them to get credit for their work. Select 'Yes' below to register your reviewing activity at Web of Science Reviewer Recognition Service (formerly Publons); note that the content of your review will not be visible on Web of Science.

No

We sincerely thank the reviewers for their evaluation of our manuscript. We greatly appreciate their insightful feedback, which has helped us refine our conclusions and strengthen the study. We are particularly grateful for the reviewers' recognition of key aspects of our work:

Reviewer #1 noted that our "*experimental design and analyses are adequate.*" Reviewer #2 described our study as "*an interesting and logical series of studies that investigate the activity and signaling of the GPCR TBXA2R in TNBC cells*" and highlighted that our "*data include orthogonal assays with similar results, and the manuscript is written clearly and the data are displayed well.*" Similarly, Reviewer #3 emphasized that our study "*is well-conceived and executed, and the results are clearly described and presented in the manuscript.*" We also appreciate Reviewer #4's positive remarks that "*the authors show an interesting and conclusive work on the activation of ERM proteins upon TBXA2R signaling*" and their recognition of the "*elegant use of the ebBRET biosensor to assess ERM-protein activation.*"

All modifications made in response to the reviewers' comments are highlighted in blue throughout the revised manuscript.

We now provide a detailed response to each reviewer's comments below.

Reviewer #1

Experimental design and analyses are adequate, even though certain experiments lack appropriate controls or employ the wrong statistical tests. However, the study primarily relies on a single TNBC cell line and heavy use of overexpression systems and/or small molecule inhibitors, raising concerns about the generalizability and specificity of the findings. Furthermore, several conclusions appear premature and unsupported by the current data. Critical controls and additional validation experiments are necessary to support the claims about the role of TBXA2R in metastasis and to justify the strong mechanistic conclusions drawn.

We thank Reviewer #1 for their thoughtful and critical evaluation of our manuscript. We appreciate the opportunity to clarify our approach and to strengthen the manuscript in response to the concerns raised.

In the revised version, we have added new data, controls and new statistical analyses where appropriate. We also have introduced critical clarifications and tempered certain conclusions to ensure they are fully supported by the presented evidence.

To address the concern about reliance on a single TNBC model, we now include new experiments performed in MDA-MB-231 cells in addition to Hs578T cells. These experiments show that TBXA2R activation similarly enhances cell motility in a manner dependent on ERM activation by SLK/LOK (see new Fig. 5H, I, K), thereby reinforcing the generalizability of our findings.

More broadly, the revised manuscript now integrates signaling analyses across six TNBC cell lines, complemented by mechanistic studies in HEK293T cells, in vitro functional assays in two representative TNBC models (Hs578T and MDA-MB-231), and in vivo experiments using Hs578T cells. Together, these data provide a robust and coherent framework supporting the role of the TBXA2R–ERM signaling axis in promoting TNBC cell motility, invasion, and metastatic dissemination.

Figure 1

TBXA2R expression should be shown to understand whether different ebBRET signals are dependent on the overexpression levels of TBXA2R.

When different conditions were compared, cells were reverse transfected in suspension before splitting and plating, ensuring that all conditions expressed similar TBXA2R levels within the same experiment. This is now indicated in the material and method section.

E-F: As ERM levels change over time, one would like to understand whether this is due to misloading or whether there is an underlying biological event going on in the stimulated cells. Are total ERM levels really changing over time? Please add a blot for 1-2 housekeeping proteins as loading controls. This is also crucial to clarify the kinetics of ERM activation; such notable intensity variations make quantifications of non-linear WB signals not fully reliable. In F, mean and SD should be plotted.

We thank the reviewer for this constructive comment and have addressed all aspects raised. We now include actin blots as a housekeeping control to confirm consistent loading across all conditions and to normalize pERM levels in Fig. 1E–F, which removes the variability previously observed when using total ERM. Actin controls were added whenever total ERM signals fluctuated noticeably across conditions, and panel F has been updated to display mean \pm SD, as recommended.

In response to the reviewer's comment, we chose to normalize to actin because total ERM is not always a reliable loading control under conditions that modulate ERM phosphorylation. The apparent variability in total ERM signal over time most likely reflects the biochemical properties of the antibody rather than unequal loading. The total ERM polyclonal antibody was generated against a peptide corresponding to residues surrounding Thr567 of human ezrin, a region highly conserved in moesin and radixin, whereas the same sequence carrying the phosphorylated residue was used to raise the pERM antibody. Phosphorylation on this regulatory threonine (Thr567 in ezrin, Thr558 in moesin, and Thr564 in radixin) alters epitope accessibility, thereby affecting antibody recognition efficiency during activation, particularly at peak phosphorylation. This property can thus introduce apparent non-linearity in the total ERM signal.

G: The authors need to use a PM marker if they want to claim that pERM increases at the cell cortex. TBXA2R localization should also be shown.

We and others have repeatedly demonstrated that the primary site of ERM phosphorylation is the plasma membrane (PMID: 20308985; PMID: 35482006). Due to how we quantified signal intensity (manual tracking), Figure 1H reports signal intensity at **the cell periphery**, not specifically the plasma membrane.

In addition, the BRET-based ERM biosensors used in Figures 1 and 2 are specifically designed to report conformational activation of ERMs at the plasma membrane. This is achieved by targeting the biosensor to the inner leaflet of the plasma membrane via a myristoylation (Myr) signal followed by a polybasic (PB) motif (characterized in PMID: 33712451 and PMID: 35482006). The strong BRET signal upon TBXA2R stimulation confirms ERM activation at this compartment. Taken together and given the extensive literature supporting ERM phosphorylation at the cortex, we believe that additional PM labeling is not necessary to confirm this localization

Regarding TBXA2R localization, we agree that direct visualization provides useful complementary information. Unfortunately, we were unable to identify an antibody that reliably detects endogenous TBXA2R by immunofluorescence. Using confocal microscopy, we expressed a GFP-tagged version of TBXA2R, which localized predominantly to the plasma membrane, with some intracellular signal also detected, an expected feature in overexpression systems reflecting receptor trafficking and biosynthetic pools (Fig 1 for reviewer). Additionally, the ability of extracellular U46619 to trigger downstream signaling provides functional evidence that TBXA2R is correctly positioned at the plasma membrane. Because there was no *a priori* reason to believe that the TBXA2R would not be localized at the cell surface, we did not include this figure in the revised manuscript and only provide it for the perusal of the reviewer. If the editor and reviewer think it should be included, we could add it as a supplementary data.

Figure 1 for reviewers: Hs578T cells stably expressing TBXA2R-GFP. TBXA2R-GFP is enriched at the cell periphery, indicating that the receptor properly localizes to the plasma membrane when overexpressed.

Figure 2

A: This reviewer cannot see the purported partial inhibition in Ga12/13 KO cells. Are differences between the two KOs significant? Furthermore, there are reports indicating that YM-254890 may not be specific for Gαq. Experiments on double KO cells are needed to assess the possible redundancy between the two Gα subfamilies.

The selectivity of YM-254890 for Gαq/11 has been extensively characterized, including in Figure 1 of *PMID: 26658454*, where its specificity to Gαq/11 was rigorously validated and largely used by the community as a selective inhibitor. We have also used YM-254890 in several previous studies and have not observed off-target effects at the concentration used in our current experiments.

To directly address the reviewer's comment on redundancy between Gαq/11 and Gα12/13, we performed additional experiments using a HEK-293 Gα protein-null cell line (lacking all major Gα subunits), in which we reintroduced Gαq or Gα12 individually. These new data show that re-expression of either Gαq or Gα13 is sufficient to restore ERM activation upon TBXA2R stimulation, supporting the idea that these two G protein subfamilies act redundantly in this context. These results are now included in the revised manuscript (Fig 2B) and discussed on page 8 lines 12-21.

Finally, as requested, we performed additional statistical analyses on Figure 2A, which confirmed a significant difference between control cells and either $G\alpha q/11$ -KO or $G\alpha 12/13$ -KO cells in Ezrin activation following U46619 stimulation. These analyses also revealed a difference in the extent of Ezrin opening between $G\alpha q/11$ -KO and $G\alpha 12/13$ -KO cells upon U46619 treatment. Although direct comparison between these engineered CRISPR cell lines may be of limited interpretive value, the key observation is that when $G\alpha 12/13$ is knocked out **and** $G\alpha q/11$ is inhibited by YM-254890, Ezrin no longer undergoes opening in response to U46619. We therefore rephrased our conclusions in page 8, line 7-8 and 12-21.

C-D: it is important to add a positive control for the activity of Y-27632 in these experiments. Please show that a ROCK-dependent effect is inhibited in the treated cells.

G: The working model is premature as it is unknown whether ROCKi was active. While asking for ROCK1/2 KO cells would be too much, this claim is far-fetched.

The effectiveness of the ROCK inhibitor Y-27632 in HEK293T cells has already been previously demonstrated in our earlier work (*PMID: 35482006*, Fig. S2A-B), where we showed that Y-27632 treatment abolishes phosphorylation of MLC2, a well-established ROCK substrate. We now cite this reference in the revised manuscript to support the inhibitor's efficacy.

As suggested, we also include new data demonstrating that Y-27632 effectively inhibits ROCK activity in Hs578T cells. Specifically, we show that U46619 stimulation induces robust phosphorylation of MLC2, confirming activation of Rho-ROCK signaling downstream of TBXA2R. Importantly, this phosphorylation is abolished by Y-27632 treatment, confirming that the compound inhibits ROCK kinases under our experimental conditions (Fig 4I,J). These data are now described on page 11, in the 4 last lines.

Despite the confirmed inhibition of ROCK kinases in both HEK293T and Hs578T cells, Y-27632 has no effect on ERM phosphorylation following TBXA2R activation. In contrast, ERM phosphorylation is suppressed upon depletion, knockout and chemical inhibition of SLK and LOK. These findings now support our model and the conclusion that in Hek293T and Hs578T cells, ERM activation downstream of TBXA2R does not depend on ROCK but instead requires SLK and LOK.

Figure 3

B: In the legend, it is not clear what grey and light red colours mark.

In the revised version, we have updated the figure legend to clearly indicate the meaning of the grey and light red colors. Specifically, light red represents cells treated with U46619, while grey represents cells treated with the vehicle. These clarifications have been added to ensure the figure is self-explanatory.

E-F: This reviewer finds it difficult to believe that p-ERM and TBXA2R signal intensities at the cell cortex could be reliably quantified using IHC images. The representative samples would indicate that p-ERM and TBXA2R positivity are not correlated. It would be crucial to show examples for each of the TNBC subgroups the existence of which is inferred based on p-ERM and TBXA2R staining.

The conclusion that "no TNBC samples exhibited high TBXA2R expression and low levels of p-ERMs, further supporting a role for TBXA2R signalling in ERM activation in TNBC" is an overstatement.

We agree that signal quantification at the plasma membrane in IHC samples can be unreliable due to variable preservation of membrane structures across tissue sections. For this reason, we did not attempt

to quantify p-ERM and TBXA2R specifically at the cell cortex. As indicated in the original manuscript, our goal was to assess overall cellular levels of phosphorylated ERMs and TBXA2R in TNBC samples using tissue microarrays (TMAs), not subcellular localization. To clarify, our statement in the manuscript referred to general expression patterns observed in the TMA and not to precise co-localization at the plasma membrane.

We acknowledge that the conclusion stating “no TNBC samples exhibited high TBXA2R expression and low levels of p-ERMs, further supporting a role for TBXA2R signaling in ERM activation in TNBC” may overstate the strength of this association, especially considering the modest size of the TMA cohort (55 cases). We have revised this sentence in the manuscript to more cautiously reflect the descriptive nature of the analysis and the limitations inherent to IHC-based quantification:

‘Notably, none of the TNBC samples exhibited high TBXA2R expression alongside low p-ERM levels. While the sample size is limited, this is consistent with a functional link between TBXA2R expression and ERM activation.’

In response to the reviewer’s suggestion, we now include representative images from each of the identified TNBC subgroups in the revised figure to illustrate the range of p-ERM and TBXA2R staining patterns observed (Fig 3E).

Figure 4

The authors wrote that “We focused on the Hs578T cell line, which showed a median level of TBXA2R mRNA expression among the six TNBC cell lines tested”. I do not understand the rationale for it as anti-TBXA2R antibodies detecting endogenous TBXA2R are available and thus why not use the median protein levels?

Unfortunately, the available TBXA2R antibody only works in immunohistochemistry and is not suitable for immunofluorescence or Western blot. Therefore, we based our rationale and analysis on RNA expression data to select cell lines expressing different levels of TBXA2R. This clarification was added in the revised manuscript (page 10 lines 20-26):

‘In the absence of an antibody that reliably detects endogenous TBXA2R in cultured cells by Western blot or immunofluorescence, we selected six TNBC cells lines based on TBXA2R mRNA expression levels reported in the Cancer Dependency Map Portal (Tsherniak et al., 2017) (Fig S2A).’

Figure 5

Effects of the knockouts are subtle, and rescue experiments would be needed to corroborate these results. The employed statistical analysis is prone to overestimating differences. The authors should use the superplots instead.

To address the reviewer’s concern regarding statistical representation, we have reanalyzed the relevant datasets using superplots, which are now included in the revised figures. These additions strengthen our conclusions and better reflect the variability and reproducibility of our experiments across biological replicates. Moreover, inhibition of Moesin by two independent sgRNAs (Fig 5D, 5M) or siRNAs (Fig S3G)

consistently reduced cell velocity by more than 25% upon U46619 stimulation. A comparable reduction was observed when SLK/LOK activity was inhibited, either pharmacologically using Cpd31 in Hs578T (Fig 5J) and MDA-MB-231 (Fig 5K) or genetically by CRISPR knockout (Fig 5E and 5N). Thus, the observed effects are reproducible across complementary approaches and experimental systems, indicating that they are not subtle but robust and biologically meaningful.

The authors might also decide to use other TNBC cell lines to explore the functional relevance of this pathway in BC progression. This is particularly important because Hs578T are poorly tumorigenic, and they often do not form palpable tumours in mice.

And Figure 6

The fact that Hs578T are poorly tumorigenic in mice is likely the reason why the authors used the experimental metastasis model. However, it is puzzling that metastases were studied in the liver but not in the lungs.

As the reviewer pointed out, a recent study (PMID: 38468326) reported that Hs578T cells have low tumorigenic potential in an orthotopic injection model. However, it is well established that tumorigenicity and metastatic potential of cell lines vary across laboratories due to differences in passage history, culture conditions, experimental setups, or selective pressures. For instance, this variability was documented in the case of MCF7, a breast cancer cell line widely used in tumorigenic assays (PMID: 30089904).

Importantly, several independent studies have shown that Hs578T cells can form palpable primary tumors (see Fig 4F in PMID: 36625722) and can metastasize to the lungs and liver after tail vein injection (see Figure 6 of PMID: 31351450), supporting their use in metastatic models. The Hs578T cell line used in our study has been authenticated by STR profiling at ATCC and we have consistently observed robust tumor formation after both ectopic and orthotopic injection into immunocompromised NSG mice. In our hands, these tumors also metastasize to the lungs and liver. While our initial manuscript primarily focused on liver metastases, we also observed increased tumor burden in the lungs upon TBXA2R activation, which was fully reversed by MSN knockout. Due to extensive metastatic burden in U46619-treated lungs, IHC-based quantification was not reliable. However, we now include macroscopic quantification of lung metastases in the revised manuscript (Fig 6B), and discuss the data on page 14-15.

Taken together, these findings support the use of Hs578T cells as a relevant and validated model for studying TBXA2R-driven metastasis in TNBC.

Furthermore, the whole approach is rather artefactual as the TBXA2R agonist was administered for the entire duration of these experiments. What is the pathological relevance of such a study? Including a spontaneous metastasis model or alternative TNBC lines that mimic human disease more closely would help strengthen the functional relevance of this pathway in BC progression and study's translational relevance.

The *in vivo* experiments were designed to test the relevance of our *in vitro* findings and evaluate whether sustained activation of the TBXA2R–ERM signaling axis can promote metastatic colonization *in vivo*.

To mimic a pathological context in which thromboxane A2 (TXA2), the natural ligand of TBXA2R, is chronically elevated, as documented in several cancers (PMID: 22296266), we chose to continuously

administer a stable TBXA2R agonist throughout the experiment. This strategy reflects the biological scenario in which TXA2, a short-lived prostanoid (half-life ~30 seconds), is persistently produced by activated platelets, endothelial cells, macrophages, neutrophils, or tumor cells within the tumor microenvironment and metastatic niches (PMID: 36234768). This rationale has now been more clearly explained in the revised manuscript on page 14, lines 15-21.

While we acknowledge that the tail-vein injection model bypasses early steps of the metastatic cascade, it is a well-established and widely accepted approach to specifically assess late-stage events such as extravasation, survival, and metastatic outgrowth. Given our focus on evaluating the sufficiency of TBXA2R–ERM signaling to drive these late steps, we believe this model is well suited to the biological question at hand. Our approach is aligned with prior studies of TBXA2R in cancer metastasis (e.g., PMID: 30907747), and our in vivo findings are in strong concordance with our in vitro results, collectively supporting the role of this signaling axis in TNBC dissemination.

We recognize the value of additional models that more fully recapitulate the metastatic cascade and tumor-host interactions. However, such studies would require significant time and resources and would delay the dissemination of our mechanistic insights. We have now explicitly noted in the Discussion that this work establishes a conceptual and experimental framework for future investigations using orthotopic or spontaneous models to further define the contribution of TBXA2R signaling to TNBC progression Page 19 lines 1-7.

Figure S2 B-M: the pERM signal appears to be perinuclear in some of the tested cell lines. Please use a PM marker.

Please see answer to Fig 1G

Figure S3 The authors should use the superplots to analyse the cell migration data.

This was done in the revised manuscript.

Discussion

The claim that "our findings demonstrated that kinases of the SLK family are the only kinases needed for ERM activation by TBXA2R" should be tuned down as only 2 cell lines were tested. In this section, the authors should also discuss the proposed pro-metastatic functions of TXA2 and TXA2R in more detail, including vascular permeability. The sweeping conclusion that "TBXA2R expression correlates with phosphorylation and activation of ERMs in TNBC patient samples" clashes with the authors' own results; please stick to the data.

In the revised discussion, we have toned down the statement that SLK family are the only kinases needed for ERM activation by TBXA2R in the cells tested. to reflect the fact that our conclusions are based on functional studies done only in now three different cell lines (HEK293T, Hs578T and MDA-MB-231). We now state that our findings support a predominant role for SLK family kinases in this context, while acknowledging the possibility that other kinases may contribute in different cellular settings:

'While it is currently unclear whether alternative mechanisms proposed for other GPCRs, such as the involvement of GRK2 (Cant and Pitcher, 2005) or direct binding of ERM to $G\alpha_{13}$ (Vaiskunaite et al., 2000) contribute to ERM activation downstream of TBXA2R, our findings indicate that SLK and LOK play a predominant role in mediating ERM activation in this context. This conclusion is based on functional studies conducted in three different cell lines and does not exclude the possibility that other kinases may contribute in distinct cellular or physiological settings'. (see page 17 lines 18-21)

Second, we appreciate the suggestion to further elaborate on the known pro-metastatic functions of thromboxane A2 and its receptor. We have expanded the Discussion to include a brief overview of the literature describing TBXA2R's roles in promoting vascular permeability, platelet aggregation, and inflammation, all of which may facilitate metastasis. Relevant references have been added to support this broader pathological context (page 18 lines 12-20):

'In addition to its role in cancer cell-intrinsic signaling, TXA_2 -TBXA2R signaling has also been implicated in several non-cell-autonomous pro-metastatic processes. TXA_2 has been shown to increase vascular permeability, promote platelet aggregation, and enhance inflammatory cell recruitment, different factors that collectively could facilitate tumor cell extravasation and metastatic seeding (Ashton et al., 2022; Liao et al., 2023; Nie et al., 2000; Xue et al., 2024). Our findings complement these non-cell-autonomous effects by indicating that TXA_2 -TBXA2R signaling may contribute to metastasis both by directly enhancing tumor cell motility via ERM activation and by shaping a permissive metastatic niche.'

Finally, we agree that the original wording regarding the correlation between TBXA2R expression and ERM phosphorylation in TNBC patient samples may be too strong. In the revised text, we have rephrased this sentence to more accurately reflect our data, emphasizing that we observed a pattern in the TMA cohort where high TBXA2R expression was not seen in tumors with low p-ERM staining (page 18, lines 24-27):

'Overexpression of essential components of this pathway could upregulate this TBXA2R-ERM signaling axis. Supporting this notion, we observed in our TMA cohort that high TBXA2R expression was not detected in tumors with low levels of phosphorylated ERMs, a pattern consistent with a link between TBXA2R signaling and ERM activation.'

Reviewer #2

Leguay et al present an interesting and logical series studies that investigate the activity and signaling of the GPCR TBXA2R in TNBC cells. The premise of the overall study is that metastasis is often associated with a more invasive/motile cancer cell phenotype. The investigators have an interest in ERM (Ezrin, Radixin, Moesin) proteins, which have been implicated in cell motility. The authors link stimulation of TBXA2R, a GPCR, to activation of ERM proteins and also show that TBXA2R is associated with worse outcome in TNBC patients. Through the use of genetic and pharmacologic tools the authors provide convincing biochemical and cell based data to support their model that stimulation of TBXA2R activates $G\alpha_{11}$ & $G\alpha_{12/13}$ which subsequently stimulate RhoA and SLK/LOK which then phosphorylate ERMs. The authors show relevant biologic consequences of the pathway. Data include orthogonal assays with similar results and the manuscript is written clearly and the data are displayed well. Overall it is a solid story that is largely well done. There are a few comments that should be addressed.

We sincerely thank the reviewer for their positive evaluation of our work and for recognizing the logical flow and scientific rigor of our study. We appreciate the acknowledgment that our study presents a solid and well-constructed investigation into TBXA2R signaling in TNBC cells and that our data are convincing, well-controlled, and clearly presented. We are also grateful for the recognition of the relevance of our findings, particularly in linking TBXA2R activation to ERM phosphorylation, RhoA-SLK/LOK signaling, and the biologic consequences of this pathway in TNBC progression. We acknowledge the reviewer's suggestions for improvement and have carefully addressed their comments in our point-by-point responses.

1. All the biochemical/cell based in vitro data exploit the use of small molecule agonists of TBXA2R, not the natural ligand. A comment on this and why use of TXA2 is not feasible would be helpful to the reader.

TXA2 is chemically unstable, with a biological half-life of approximately 30 seconds (*PMID: 18374420*). This instability makes it impractical for use in biochemical, cell-based and *in vivo* assays, which is why we opted for a synthetic agonist to ensure reproducible and reliable activation of TBXA2R. We included this explanation in the revised manuscript on page 6 lines 20-22 to clarify our choice for the reader:

'Given the very short half-life of thromboxane A2 that makes its use impractical in many assays, we used the synthetic agonist U46619, a stable, selective and reliable tool to activate TBXA2R in cellular assays (Coleman et al., 1981).'

2. The data in figures 1-5 are solid and clear. However, I suggest adding a higher magnification inset for the IHC images shown in Fig 3E. It would be useful to be able to distinguish cells in the IHC, a higher mag shot should suffice.

We appreciate this positive evaluation. We also agree that a higher magnification inset would improve the visualization of individual cells in the IHC images. This was addressed in the revised manuscript in the new Fig. 3E.

3. A) The use of Hs578t cells for the in vivo modeling is unfortunate. Additionally, the use of iv injection to in a study focused on cell invasion is also unfortunate. The metastatic propensity of Hs578t is not clear, in fact a recent report comparing metastasis in breast cancer cell lines shows that Hs578t perform poorly in terms of metastasis after orthotopic injection (see *PMID 38468326*). I searched the literature a bit to try and find other examples of iv injection of Hs578t cells, I found 1 (*PMID:27654855*, I did not search exhaustively), this paper shows significant lung metastasis and does not mention liver metastases. Were other breast cancer cells investigated for the in vivo studies?

As we answered to reviewer 1, the conclusion that Hs578T cells are poorly tumorigenic appears to be based on a single study (*PMID: 38468326*). However, it is well established that tumorigenicity and metastatic potential of cell lines vary across laboratories due to differences in passage history, culture

conditions, experimental setups, or selective pressures. This variability was documented in the case of MCF7, a breast cancer cell line widely used in tumorigenic assays (*PMID: 30089904*).

Importantly, several independent studies have shown that Hs578T cells can form palpable primary tumors (for example see Fig 4F in *PMID: 36625722*) and metastasize to the lungs and liver after tail vein injection (for example see Figure 6 of *PMID: 31351450*), supporting their use in metastatic models. The Hs578T cell line used in our study has been authenticated by STR profiling at ATCC and we have consistently observed robust tumor formation after both ectopic and orthotopic injection into immunocompromised NSG mice. In our hands, these tumors also metastasize to the lungs and liver.

While our initial manuscript primarily focused on liver metastases, we also observed increased tumor burden in the lungs upon TBXA2R activation, which was fully reversed by MSN knockout. Due to extensive metastases in U46619-treated lungs, IHC-based quantification was not reliable. However, we now include macroscopic quantification of lung metastases in the revised manuscript (Fig 6B) and describe the results on page 14 lines 22-29.

While we acknowledge that the tail-vein injection model bypasses early steps of the metastatic cascade, it is a well-established and widely accepted approach to specifically assess late-stage events such as extravasation, survival, and metastatic outgrowth. Given our focus on evaluating the sufficiency of TBXA2R–ERM signaling to drive these late steps, we believe this model is well suited to the biological question at hand. Our approach is aligned with prior studies of TBXA2R in cancer metastasis (e.g., *PMID: 30907747*), and our *in vivo* findings are in strong concordance with our *in vitro* results, collectively supporting the role of this signaling axis in TNBC dissemination.

We recognize the value of additional models that more fully recapitulate the metastatic cascade and tumor-host interactions. However, such studies would require significant time and resources and would delay the dissemination of our mechanistic insights. We have now explicitly noted in the Discussion that this work establishes a conceptual and experimental framework for future investigations using orthotopic or spontaneous models to further define the contribution of TBXA2R signaling to TNBC progression Page 19 lines 1-7.

B) Why I was interested is because the typical organ that is seeded post iv injection is the lungs (as seen in the above ref), liver metastases post iv injection are not common, especially with breast cancer cells. What did the lungs look like in your experiments?

Regarding the site of metastases, we indeed observed metastases in the lungs. Our macroscopic quantification showed a similar trend of increased tumor burden in the lungs and liver after U46619 treatment, which was fully reversed by MSN knockout. However, due to the extended treatment duration, the lungs in the U46619-treated group were heavily metastasized, preventing reliable quantification by IHC.

C) Further while the data presented in figure 6 are supportive of the overall conclusions, the data is modest at best in terms of metastatic burden. Repetition of the experiment using a breast cancer cell line injected orthotopically would likely be more useful in highlighting the importance of the pathway to

metastasis. I understand performing an orthotopic assay may be outside the scope of the study, but it would provide greater impact given the focus of the paper on cell invasion.

We appreciate the reviewer's recognition that the data in Figure 6 support our overall conclusions. We agree that performing an orthotopic injection model could further strengthen the study by providing additional insights into the role of the TBXA2R-ERM pathway in metastasis. However, conducting these additional experiments is beyond the scope of the current study and would significantly delay the publication of our findings especially regarding the novel mechanistic insights. Given that our in vivo data are consistent with our in vitro results, we believe that the current dataset provides a solid foundation for understanding the contribution of this pathway to TNBC progression and open the path to future studies.

Reviewer #3

The Ezrin, radixin, and moesin (ERM) family of proteins orchestrate morphological changes that potentiate metastatic invasion in cancer cells. In this study, Leguay et al. identify the GPCR, TBXA2R, as a key activator of the ERM proteins which promotes motility and invasion in triple-negative breast cancer (TNBC) cells. Using BRET-based sensors developed by them previously for monitoring the activation of ERM proteins and building upon their previous findings on the role of the small GTPase RhoA in the activation of ERM proteins, the authors carefully dissect the molecular pathway leading to the activation of ERM proteins upon stimulation of the TBX2AR. The authors also establish the pathological relevance of the pathway in TNBC using in vitro and in vivo models, opening up possibilities for targeting this pathway in cancer cells. Overall, the study is well-conceived and executed, and the results are clearly described and presented in the manuscript. However, the following comments must be addressed before publication.

We thank the reviewer for the positive evaluation of our study. We appreciate the recognition of our work in dissecting the TBXA2R-ERM signaling pathway and its relevance to TNBC progression. We are glad that the clarity and execution of our experiments were well received. Below, we address the specific comments raised by this reviewer.

Fig 1C - Why p-ERM was normalized over Ezrin and not ERM? It would be more appropriate and consistent to normalize against the ERM signal as done in other experiments in the manuscript.

Fig 1E and S3C - The levels of total ERM also seem to change with increasing treatment times. This must be clarified and discussed in the manuscript.

As we previously addressed in our response to Reviewer #1, we agree that the observed variation in total ERM signal over time could reflect either differences in protein loading or genuine biological changes in ERM protein levels. We have now included blots for the Actin housekeeping protein as a loading control to assess this. These controls confirm that the loading was consistent across time points, suggesting that the variations in total ERM signal are not due to misloading.

It is indeed expected that total ERM antibodies detect both phosphorylated and non-phosphorylated forms. However, we have noted that ERM phosphorylation may alter epitope accessibility or protein

conformation, potentially affecting antibody recognition and thereby introducing variability in the total ERM signal. This could contribute to non-linear behavior in the quantification, particularly during peak activation. To address this, we have now normalized pERM levels to Actin rather than to total ERMs in our revised quantifications, providing a more reliable readout of activation kinetics.

Fig 1F - Why is the mean of all three independent experiments not presented here as in S3C?

3 independent experiments are now presented in new Fig 1F.

Fig 2E - Though SLK seems to play a dominant role in the phosphorylation of ERM in HEK293T cells, the depletion of LOK also substantially reduces the phosphorylation of ERM in the representative figure (Fig 2E), which is not reflected in the quantification (Fig 2F). Indeed, both SLK and LOK seem to be equally crucial in Hs578T cells (Fig 4I), unlike the conclusion here. The authors must check if the quantifications were affected by any white spots in the blot for total ERM as seen in the representative figure. If necessary, the authors must include additional replicates, and the model in Fig 2G should be updated accordingly. If the contributions of LOK are indeed quite minimal in HEK293T cells, then the difference in Hs578T cells must be adequately highlighted and discussed rather than broadly mentioning similar results were observed in both cell lines.

As previously mentioned to reviewer 1, one potential drawback of our quantification is that the ERM antibody also recognizes a fraction of phosphorylated ERMs. To ensure consistency across all experiments, actin controls were added whenever total ERM signals fluctuated noticeably across conditions,

Additionally, we performed RT-qPCR analysis in HEK293 and Hs578T cells, which revealed that while LOK and SLK mRNA levels are comparable in HEK293 cells, LOK is more highly expressed than SLK in Hs578T cells. This difference in expression levels could explain the variation in signaling between the two cell lines. This is now discussed in page 12 line 1-10.

The discussion mentions that SLK kinases are the only kinases needed for ERM activation, which conflicts with findings from Hs578T cells, where both SLK and LOK contribute to ERM phosphorylation (Fig 4I). The authors should revise this to reflect their data accurately.

We apologize for the unclear wording. In referring to SLK kinases, we intended to include both SLK and LOK. This is corrected in the revised manuscript to accurately reflect our data:

leading to ERM activation via kinases of the SLK/LOK family

FigS3B should cite the source dataset and not just the database. Also, details of how the extracted data was processed (if any) should be described clearly.

When multiple treatments are involved (for, e.g. U46619 and staurosporine), the exact sequence of treatments and the overlap in timings of different treatments must be clearly mentioned. E.g. fig 1A and 1C.

There are a few grammatical errors which need to be fixed. E.g. Paragraph 2 in the second section of results - We next aimed to identify (not identifying) which kinase(s) acts downstream of TBXA2R.

The grammatical errors are corrected in the revised version. Legend of FigS3B has been corrected accordingly and a paragraph for multiple treatment has been added in the material and methods.

Reviewer #4

Overall, the authors show an interesting and conclusive work on the activation of ERM proteins upon TBXA2R signaling. The use of the eBBRET biosensor to assess ERM-protein activation enables elegant investigation of activation modalities. The Thromboxane A2 analogue U46619 robustly shows activation of ERM proteins in eBBRET assays as well as an increase in ERM-protein phosphorylation status. The functional effects of this signaling pathway are shown convincingly for moesin, where moesin mediates an TBXA2R mediated increase in cell motility, invasion and metastasis of triple-negative breast cancer Hs578 cells in vitro and in vivo. Nonetheless, some points need to be clarified.

Thank you for this positive evaluation of our work and for recognizing the strength of our approach, including the use of eBBRET biosensors and the functional relevance of TBXA2R-mediated ERM activation. We appreciate this constructive feedback and address the points raised by this reviewer in the responses below.

Comment 1: In the title the authors state, that ERM-activation via TBXA2R is controlling invasion and motility of triple-negative breast cancer cells. In the manuscript, there is only data supporting this assumption for moesin (MSN). Therefore, the authors need to change the title accordingly or support additional experiments for the other two ERM-proteins radixin and ezrin.

All three ERM proteins are activated by TBXA2R, but our data demonstrate that moesin is the main ERM in Hs578T cells due to its higher mRNA expression (Fig. S3B). To accurately reflect our findings, we replaced "ERM" with "moesin" and adjusted the title accordingly in the revised manuscript.

The G protein-coupled receptor TBXA2R activates ERMs and promotes moesin-dependent motility and invasion of triple-negative breast cancer cells.

Throughout the experiments, the p-ERM antibody is used to measure ERM-protein activation. Since the effects on invasion and motility observed in Hs578 cells are mainly mediated through moesin, it would be necessary to see, at least for one experiment per cell line (HEK293T, Hs578) the detailed phosphorylation status of ezrin, radixin and moesin separately. As there are specific, phospho-detecting antibodies for this case, this could be done rather easy. Furthermore, showing specific increase of phosphorylated moesin would support the functional data shown in Figure 5 and 6. To investigate the functional effect of TBXA2R mediated activation of ezrin and radixin on cell motility and invasion, similar experiments could be done in e.g. HMC-1-8 breast cancer cells (high ezrin expression) and HCC1187 (high radixin expression).

Unfortunately, while some companies market phospho-specific antibodies for Ezrin, Radixin, or Moesin individually, these antibodies are not truly specific and instead function as pan-phospho-ERM antibodies. This lack of specificity is due to the 100% sequence conservation around the phosphorylated threonine across all three ERM proteins. We contacted the company regarding this issue, and while they acknowledged the problem, it has not been corrected.

Given this limitation, it is not possible to distinguish the phosphorylation status of each ERM individually using currently available antibodies. Therefore, we rely on p-ERM as a readout of ERM activation in our experiments. By knocking out or knocking down each ERM individually, we demonstrated that only the loss of Moesin caused a drastic reduction in U46619-induced total ERM phosphorylation (Fig. 5A-B and S3A). This suggests that ERM phosphorylation following U46619 stimulation is primarily driven by Moesin phosphorylation.

Testing the individual contributions of Ezrin, Radixin, and Moesin to cell motility and invasion is indeed an interesting question; however, we respectfully note that such analyses fall beyond the scope of the present study.

Comment 2: Figure 1A, C, D: The concentration of staurosporine is with 100 nM relatively high for kinase inhibition. It would be informative to see the assay with increasing staurosporine concentrations, e.g. from 1 nM to 50 nM. In general, a concentration of 1-10 nM should be sufficient for kinase inhibition, preventing unspecific effects of the drug.

We agree that 100 nM staurosporine could be considered high, although many studies in the literature have used concentrations ranging from 100 nM to 10 μ M in HEK293T and other cell lines. While some degree of non-specificity is possible, our results serve primarily as a proof-of-concept, as it is well established that ERM proteins are predominantly regulated by phosphorylation.

Moreover, to address potential non-specific effects of staurosporine, we subsequently identified SLK and LOK as the kinases responsible for ERM activation upon TBXA2R induction using orthogonal assays (CRISPR knock-out and chemical inhibition using Cpd31). This provides a more specific and targeted validation of our findings.

Comment 3: The citation for the p-ERM antibody is confusing, as there is only p-Moe used in the cited paper (Roubinet, 2011). There is a p-ERM antibody commercially available (Cell Signaling, Phospho-ezrin (Thr567)/radixin (Thr564)/moesin (Thr558) Antibody #3141). Could you clarify which antibody you are using?

The antibody we used is the one described in Roubinet (2011). Although it was initially designed to detect moesin phosphorylation in *Drosophila*, the sequence surrounding the regulatory threonine is highly conserved across mammalian ERM proteins. As a result, this antibody also recognizes the phosphorylated forms of the three mammalian ERMs (Leguay et al, J Cell Science 2021 and Leguay et al, J Cell Biol 2022.)

Comment 4: From the inhibitor experiments using C3 transferase toxin (Figure 2), the authors conclude that RhoA plays a role in TBXA2R mediated ERM activation. As mentioned in the manufacturer's description, C3 toxin is inhibiting RhoA, RhoB and RhoC. Therefore, it would be necessary to repeat those

experiments under RhoA knockdown conditions (e.g. using an siRNA-based approach) to state that specifically RhoA is involved.

The reviewer is correct that C3 transferase toxin inhibits RhoA, RhoB, and RhoC, and therefore, our conclusion should not specifically attribute the effect to RhoA alone. To reflect this, we revised our conclusion and referred to Rho family GTPases instead of RhoA specifically in the revised manuscript.

Comment 5: To assess, if the findings in Figure 5 and 6 are due to the higher moesin expression in Hs578 cells or are linked to a specific function of moesin, a re-expression experiment would be informative. To achieve this, the 2D and 3D migration experiments could be redone after re-expression of moesin, ezrin and radixin separately in moesin knockdown conditions.

As previously mentioned, we acknowledge that investigating the specific functions of Ezrin, Radixin, and Moesin would be an interesting avenue of research. However, we believe that this question falls beyond the scope of the current study.

February 6, 2026

RE: Life Science Alliance Manuscript #LSA-2025-03601-T

Prof. Sébastien Carréno
Université de Montréal
Pathology and Cell Biology
Marcelle-Coutu Pavilion - room 3306-5 C.P. 6128 Succursale Centre-Ville
Montreal, Quebec H3C3J7
Canada

Dear Dr. Carréno,

Thank you for submitting your revised manuscript entitled "The G protein-coupled receptor TBXA2R activates ERMs and promotes moesin-dependent motility and invasion of triple-negative breast cancer cells". This manuscript was returned to the original referees at Review Commons whose comments are below.

As you will see, all reviewers find their previous comments have been addressed. However Reviewers 1 and 3 both express concern with the new approach to protein normalization that was modified in response to their prior requests. While normalizing to actin is appropriate, we concur with Reviewer 1 that the timecourse in Fig 1 should still include blots for total ERM even if the levels vary. More broadly and in line with the request from Reviewer 3, please state clearly throughout the results section when different loading controls were used with a justification where needed. We encourage you, if possible, to also add actin controls to the two blots remarked on by Reviewer 3. Finally please address concerns on statistics and results of LOK depletion noted by these reviewers in the manner of your choosing. We would be happy to publish your paper in Life Science Alliance pending these final revisions and formatting changes necessary to meet our guidelines.

MANUSCRIPT ORGANIZATION AND FORMATTING:

To avoid unnecessary delays in the acceptance and publication of your paper, please read the following information carefully. Full guidelines are available on our Instructions for Authors page, <https://www.life-science-alliance.org/authors>

- Please upload your main manuscript text as an editable doc file.
- Please upload your figures as single files.
- Please add the X and Bluesky handles of your host institute/organization, as well as your own and/or one of the authors, in our system.
- Please add a Summary Blurb/Alternate Abstract and Category for your manuscript in our system.
- Please mark the second corresponding author in the system and please add their ORCID ID - they should have received instructions on how to do so.
- Please include specific dataset accession numbers or other identifiers for gene expression and survival data obtained from TCGA.
- Please consult our manuscript preparation guidelines <https://www.life-science-alliance.org/manuscript-prep> and make sure your manuscript sections are in the correct order
- Please add an Author Contributions section to our system.
- Please add your main and supplementary figure legends to the main manuscript text after the references section.
- It is recommended to exclude figures from the manuscript text and upload them separately.
- Please use the [10 author names et al.] format in your references (i.e., limit the author names to the first 10).
- Please add a Conflict of Interest statement to your main manuscript text.
- Please add weight to the blots in Figure 5F and H.
- Please check the images in Supplemental Figure 3. A portion of the image in panel B (U46619 condition, merged image) appears very similar to a portion of the image in panel H (vehicle condition, merged image). Kindly ensure all images in this figure are correct.

It is Life Science Alliance policy that if requested, original data images must be made available to the editors. Failure to provide original images upon request will result in unavoidable delays in publication. Please ensure that you have access to all original

data images prior to final submission.

LSA encourages authors to provide a 30-60 second video where the study is briefly explained. We will use these videos on social media to promote the published paper and the presenting author (for examples, see <https://docs.google.com/document/d/1-UWCfbE4pGcDdcgzcmiuJI2XMBJnxKYeqRvLLrLSo8s/edit?usp=sharing>). Corresponding or first-authors are welcome to submit the video. Please submit only one video per manuscript. The video can be emailed to contact@life-science-alliance.org

FINAL FILES:

The following items are required for acceptance.

The license to publish form must be signed before your manuscript can be sent to production. A link to the license to publish form will be available to the corresponding author only. Please take a moment to check your funder requirements.

Thank you for your attention to these final processing requirements. Please revise and format the manuscript and upload materials as soon as you are able.

Thank you for this interesting contribution to the literature. We look forward to publishing your paper in Life Science Alliance.

Sincerely,

Reviewer #1 (Comments to the Authors (Required)):

The revised manuscript has improved substantially. However, a few important points remain to be addressed before publication:

- Normalization of pERM

Changes in total ERM levels, which, as the authors themselves acknowledge, do occur, are important for correctly interpreting the phosphorylation data and for building a coherent working model. Please ensure that total ERM quantification across the time-course is technically reliable and shown explicitly. A simple solution would be to rerun the time-course using a total ERM antibody recognizing a different epitope, or to run two separate gels/blots, one for pERM and one for total ERM.

- Statistical analysis

The statistical reporting requires clarification and, in some cases, adjustment. The use of superplots to display migration data is appropriate; however, it is not always clear whether hypothesis testing was performed on individual tracks or on biological replicates (independent experiments). If p-values were computed using cell-level values, this constitutes pseudo-replication and may inflate significance. Please explicitly state the unit of analysis used for each test and ensure that statistical comparisons are

performed on biological replicate summaries.

In addition, migration speed is non-negative and often right-skewed. Therefore, parametric multiple comparisons based on a single pooled variance (e.g., Holm-Šidák) should be justified, particularly given the small number of biological replicates ($n = 3$).

Reviewer #2 (Comments to the Authors (Required)):

The authors have adequately addressed the queries of the prior review.

Reviewer #3 (Comments to the Authors (Required)):

Overall, many of the points raised earlier have been addressed, and the manuscript has been considerably improved, but some concerns still remain which must be addressed before acceptance (please see below)

1) In their rebuttal, the authors argue that phosphorylation of ERM may affect recognition by the total ERM antibody in a non-linear manner, making it unreliable for normalization. Accordingly, they switch to Actin based normalization in Fig 1F and Fig 5A, and include Actin blots in several additional panels to demonstrate equal loading. Despite this, in most other quantifications the p-ERM signal is still normalized to total ERM (eg Fig 2G, 3H, 4B, 4D, 4F, 4H, 5I). In addition, in some panels Actin controls are missing altogether (eg Fig 5A and Fig 5C). A consistent normalization strategy should be applied across all figures. If reliable total ERM antibodies are not available, p-ERM quantification should be normalized to a loading control such as Actin in all cases.

2) The authors continue to state that "LOK depletion has a minimal effect on ERM phosphorylation" based on Fig 2G. However, the immunoblots shown in Fig. 2F indicate a substantial reduction in ERM phosphorylation in LOK-depleted cells. This apparent discrepancy may arise because the quantification in Fig 2G is still based on normalization to total ERM signal, which as discussed above may not be appropriate.

Dear Dr. Fessenden,

Thank you for your positive evaluation of our revised manuscript entitled "*The G protein-coupled receptor TBXA2R activates ERMs and promotes moesin-dependent motility and invasion of triple-negative breast cancer cells.*" We are pleased that the reviewers find their previous comments to have been adequately addressed.

In response to the remaining points raised by Reviewers 1 and 3 and highlighted in your decision letter, we have revised the manuscript to ensure clarity and consistency in the normalization strategy used for ERM phosphorylation across all figures.

For Fig. 1A, we reverted to the original pERM/total ERM normalization, as requested, and included the corresponding pERM/actin time-course immunoblot as supplementary material. The pERM/ERM and pERM/actin analyses derive from parallel replicate time-course experiments rather than from the same membrane. Presenting the actin-normalized series in the supplementary material provides a transparent loading control while keeping the main figure focused on ERM activation kinetics.

For Fig. 5A, pERM/total ERM normalization is not appropriate, as this panel examines Ezrin, Radixin, or Moesin depletion, where total ERM levels vary by design. We therefore retained actin-based normalization.

With respect to the two panels highlighted by Reviewer 3 (Fig. 4A and 4C), actin loading controls were not performed. However, total ERM levels remain stable across these conditions and ERM phosphorylation changes are consistently observed.

More generally, we added a brief clarification to the Materials and Methods noting that the total ERM antibody may recognize predominantly non-phosphorylated ERM but also a fraction of phosphorylated ERM, potentially in a non-linear manner. Nevertheless, normalization of pERM to total ERM remains appropriate to quantify ERM activation when total ERM levels are stable or experimentally controlled. Complementary actin normalization is provided when required to document equal loading.

Concerning Reviewer 1's comment on the statistical analysis of cell migration, migration speed was first calculated for each tracked cell within an experiment, and the mean cell speed was then computed independently for each biological replicate (n = 3 independent experiments per condition). All statistical analyses were subsequently performed on these independent experiment means, which constitute the appropriate unit of biological replication. No pooling of cell-level values across experiments was performed, thereby avoiding pseudo-replication. Statistical tests were applied to replicate means, each derived from a large number of tracked cells. Accordingly, the sampling distribution of replicate means is expected to approximate normality, supporting the use of parametric comparisons. The figure legend has been revised to explicitly state that biological replicate means were used as the unit of analysis for all statistical tests.

In response to Reviewer 2's comment on the respective roles of SLK and LOK in ERM phosphorylation in HEK293T cells, we adjusted the wording to better reflect the modest but observable decrease in ERM phosphorylation upon LOK depletion.

Finally we addressed every editorial comments.

We believe these revisions address the remaining concerns and improve the transparency of the experimental analysis. We look forward to the publication of our work in *Life Science Alliance*.

SÉBASTIEN CARRÉNO, PH.D
DIRECTOR OF ACADEMIC AFFAIRS OF IRIC
FULL PROFESSOR
DEPARTMENT OF PATHOLOGY AND CELL
BIOLOGY
FACULTY OF MEDICINE
UNIVERSITY OF MONTRÉAL

March 3, 2026

RE: Life Science Alliance Manuscript #LSA-2025-03601-TR

Dr. Sébastien Carreno
IRIC - Department of Pathology and Cell Biology, Faculty of Medicine, Université de Montréal

Dear Dr. Carreno,

Thank you for submitting your Research Article entitled "TBXA2R activates ERMs to drive motility, invasion and metastatic colonization of TNBC cells.". It is a pleasure to let you know that your manuscript is now accepted for publication in Life Science Alliance. Congratulations on this interesting work.

DISTRIBUTION OF MATERIALS:

Again, congratulations on a very nice paper. I hope you found the review process to be constructive and are pleased with how the manuscript was handled editorially. We look forward to future exciting submissions from your lab.

Sincerely,
